# Imagined speech can be decoded from low- and cross-frequency intracranial EEG features

Timothée Proix [1,12 ✉], Jaime Delgado Saa[1,12], Andy Christen[1], Stephanie Martin[1], Brian N. Pasley[2], Robert T. Knight[2,3], Xing Tian [4,5,6], David Poeppel[7,8], Werner K. Doyle[9], Orrin Devinsky [9], Luc H. Arnal [10,13], Pierre Mégevand [1,11,13] & Anne-Lise Giraud [1,13]

Reconstructing intended speech from neural activity using brain-computer interfaces holds great promises for people with severe speech production deficits. While decoding overt speech has progressed, decoding imagined speech has met limited success, mainly because the associated neural signals are weak and variable compared to overt speech, hence difficult to decode by learning algorithms. We obtained three electrocorticography datasets from 13 patients, with electrodes implanted for epilepsy evaluation, who performed overt and imagined speech production tasks. Based on recent theories of speech neural processing, we extracted consistent and specific neural features usable for future brain computer interfaces, and assessed their performance to discriminate speech items in articulatory, phonetic, and vocalic representation spaces. While high-frequency activity provided the best signal for overt speech, both low- and higher-frequency power and local cross-frequency contributed to imagined speech decoding, in particular in phonetic and vocalic, i.e. perceptual, spaces. These findings show that low-frequency power and cross-frequency dynamics contain key information for imagined speech decoding.

[1] Department of Basic Neurosciences, Faculty of Medicine, University of Geneva, Geneva, Switzerland. [2] Helen Wills Neuroscience Institute, University of California, Berkeley, Berkeley, USA. [3] Department of Psychology, University of California, Berkeley, Berkeley, USA. [4] Division of Arts and Sciences, New York University Shanghai, Shanghai, China. [5] Shanghai Key Laboratory of Brain Functional Genomics (Ministry of Education), School of Psychology and Cognitive Science, East China Normal University, Shanghai, China. [6] NYU-ECNU Institute of Brain and Cognitive Science at NYU Shanghai, Shanghai, China. [7] Department of Psychology, New York University, New York, NY, USA. [8] Ernst Strüngmann Institute for Neuroscience, Frankfurt, Germany. [9] Department of Neurology, New York University Grossman School of Medicine, New York, NY, USA. [10] Institut de l'Audition, Institut Pasteur, INSERM, F-75012 Paris, France. [11] Division of Neurology, Geneva University Hospitals, Geneva, Switzerland. [12] These authors contributed equally: Timothée Proix, Jaime Delgado Saa. [13] These authors jointly supervised this work: Luc H. Arnal, Pierre Mégevand, Anne-Lise Giraud. ✉email: timothee.proix@unige.ch

Cerebral lesions and motor neuron disease can lead to speech production deficits, or even to a complete inability to speak. For the most severely affected patients, decoding speech intentions directly from neural activity with a BCI is a promising hope. The goal pursued is to teach learning algorithms to decode neural signals from imagined speech, e.g. syllables, words, and to provide feedback to the patient so that the algorithm and the user adapt to each other. This strategy parallels what is being done in the motor domain to help paralyzed people control e.g. a robotic arm[1]. One approach to decode imagined speech is to train algorithms on neural activity corresponding to articulatory motor commands produced during overt or silently articulated speech, hoping that the learned features could ultimately be transferred to patients who are unable to speak[2–5]. Although potentially interesting, this hypothesis is limited in scope as it can only apply to cases where language and cortical motor commands are preserved (such as in motor neuron disease), i.e. a minority of the patients with severe speech production deficits[6,7]. If, as in most post-stroke aphasia cases, the cortical language network is injured, other decoding strategies must be envisaged, for instance using neural signals from the remaining intact brain regions that encode speech, e.g. regions involved in perceptual or lexical speech representations such as the temporo-parieto-occipital junction, the superior temporal gyrus, and the ventral anterior temporal regions[8,9]. Exploring these alternative hypotheses requires that researchers work directly from imagined speech neural signals, even though they are notably difficult to decode, because of their high spatial and temporal variability, their low signal-to-noise ratio, and the lack of behavioral outputs[10,11]. To advance imagined speech decoding, two preliminary key points must be clarified: (i) what brain region(s) and associated representation spaces offer the best decoding potential, and (ii) what neural features (e.g. signal frequency, cross-frequency or -regional interactions) are most informative within those spaces.

Imagined speech decoding with non-invasive techniques, i.e. surface electroencephalography (EEG) or magnetoencephalography (MEG), has so far not led to convincing results, despite recent encouraging developments (vowels and words decoded with up to ~70% accuracy for a three-class imagined speech task)[12–17]. The most effective approach so far to advance toward a real "imagined speech" decoding system is based on electrocorticographic (ECoG) signals, which are currently only recorded in patients with refractory epilepsy undergoing presurgical evaluation. During the experiments, patients are typically asked to speak aloud[18,19], imagine speaking[19] or imagine hearing speech[18], and ECoG signals are recorded simultaneously. In the overt speech condition, the recorded speech acoustics is used to inform the learning algorithms about the timing of speech production in the brain[18,19]. The main state-of-the-art feature used for overt speech decoding is the broadband high-frequency activity (BHA)[20,21]. When sampled from the premotor and motor articulatory cortex[22–24], this feature permits reasonable decoding performance. However, even though patients have an intact language and speech production system[18,25], BHA features do not seem to yield decoding accuracies for imagined or covert speech equivalent to those seen with overt speech. Alternative features or feature combinations are hence needed to advance from decoding overt speech to the more clinically relevant step of decoding imagined speech[13].

The feature space being potentially unlimited, it is essential for future aphasia treatments to reduce the space of exploitable features to the most promising ones, as for prophylactic reasons intracortical sampling will have to remain as restricted as possible. Existing speech and language theories, in particular, theories of imagined speech production, constitute an essential background to target the best speech representation level(s) and associated brain regions. While the motor hypothesis posits that imagined speech is essentially an attenuated version of overt speech with a well-specified articulatory plan (much like imagined and actual finger movements share the similar spatial organization of neural activity), the abstraction hypothesis proposes that it arises from higher-level linguistic representations that can be evoked without an explicit motor plan[10,26–30]. Between these two accounts, the flexible abstraction theory assumes that the main representation level of imagined speech is phonemic, even though subjects can retain control on the contribution of sensory and motor components[26,31–33]. In this case, neural activity is likely shaped by the way each individual imagines speech[34], whether by sub-articulating, or using perceptual (phonetic) representations, among others. An important argument for the flexible abstraction hypothesis is that silently articulated speech exhibits the phonemic similarity effect (i.e. errors involving more similar phonemes are more likely), whereas imagined speech without explicit mouthing does not[26]. Altogether these theories suggest that perceptual spaces, in particular auditory/phonetics, deserve as much attention as the articulatory dimension in imagined speech decoding.

Other current theories of speech processing[35] provide important complementary information to identify the best neural features to exploit within those spaces. An influential line of research suggests that frequency features other than BHA are critical to speech neural processing and encoding[35]. Slower frequencies, in particular, the low-gamma and theta bands could underpin phoneme- and syllable-scale processes that are essential for both speech perception and production, such as the concatenation of segment-level information (phoneme-scale) within syllable timeframes. This hierarchical embedding could be operated by nested theta/low-gamma and theta/BHA phase-amplitude cross-frequency coupling (CFC) both in speech perception and production[35–39]. The low-beta range could also contribute to speech encoding as it is implicated in top-down control during language tasks[40,41]. Together with other rhythms, such as the low-gamma band, it participates in the coordination of bottom-up and top-down information flows[42–44]. These frequency-specific neural signals could be of particular importance for intended speech decoding, as focal articulatory signals indexed by BHA are expected to be notably weaker during imagined speech.

In this study, we set out to use ECoG data sampled in patients with epilepsy to explore the range of representation level(s) and neural features that could potentially be usable in future imagined speech decoding BCIs. Rather than adopting a purely neuroengineering perspective involving large datasets and automatized feature selection procedures, we used a hypothesis-driven approach assuming a role of low-frequency neural oscillations and their cross-frequency coupling in speech processing, within both perceptual and motor representation spaces. While we confirmed the advantage of BHA for overt speech decoding, imagined speech could equally well be decoded from low- and cross-frequency features along vocalic and phonetic (perceptual) spaces.

## Results

Three imagined speech experiments were carried out in three different groups of participants implanted with ECoG electrodes (4, 4, and 5 participants with 509, 345, and 586 ECoG electrodes for studies 1, 2, and 3 respectively, Fig. 1). Each group performed a distinct task, but all three studies involved repeating out loud (overt speech) and imagining saying or hearing (imagined speech) words or syllables, depending on the study (see Methods). Despite differences in the task design, we pooled together the results to explore the full range of exploitable spatial and design/

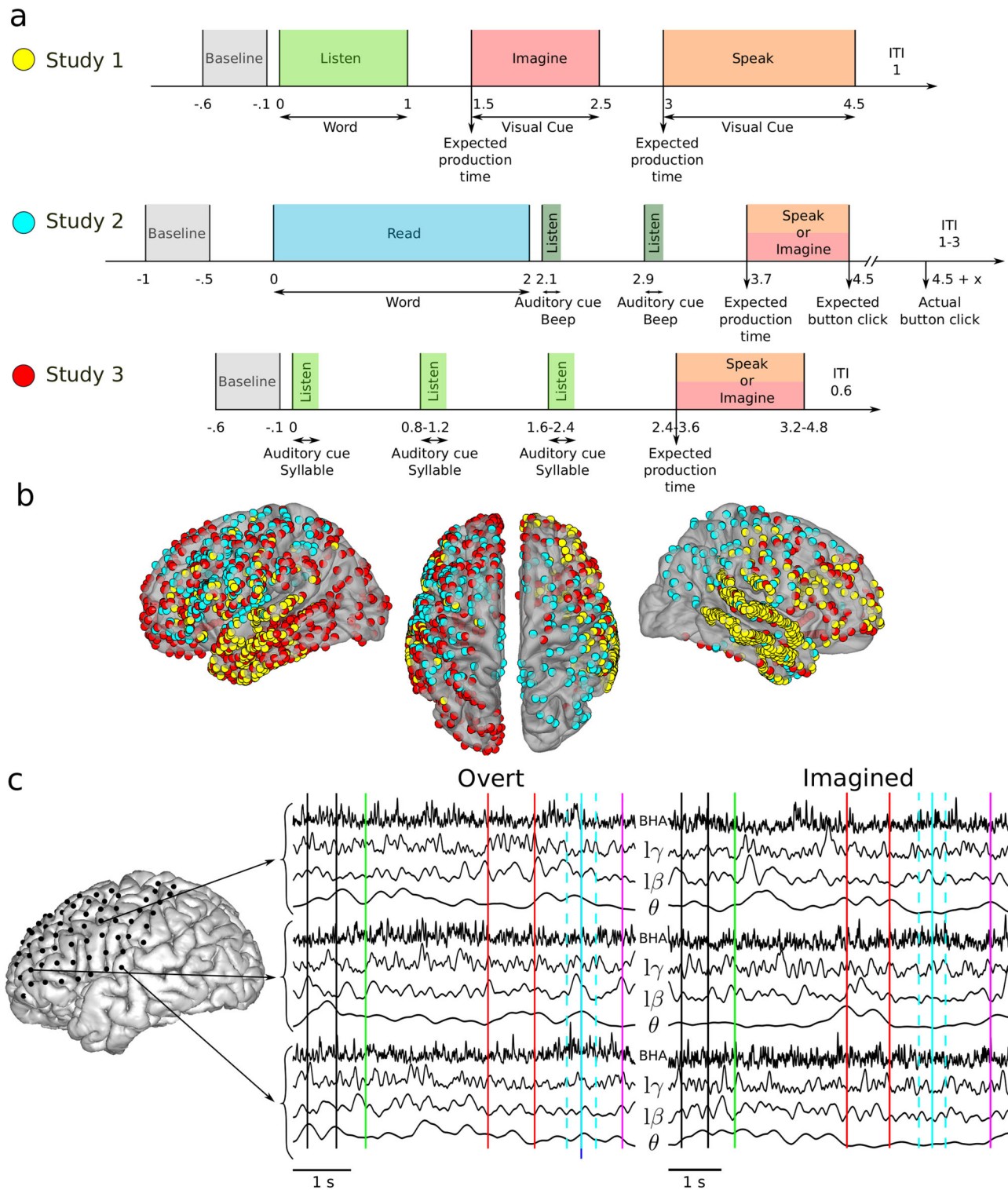

speech features. In the following, we describe compound results (across the three studies), and results for each study separately as supplementary material. ECoG signals were acquired during the experiments and preprocessed before feature extraction (Fig. 1c, Methods section).

**Speech task and item discrimination from power spectrum and phase-amplitude cross-frequency coupling**. Our primary goal was to identify if overt and imagined speech involved similar or distinct brain regions. We therefore first quantified power spectrum changes during overt or imagined speech compared to baseline for four frequency bands: theta ($\theta$, 4–8 Hz), low-beta (l$\beta$, 12–18 Hz), low-gamma (l$\gamma$, 25–35 Hz), and BHA (80–150 Hz). Overall, spatial patterns of power spectrum changes for overt and imagined speech were comparable, but not identical. Indeed, power changes were less pronounced for imagined than overt speech, as previously found[18,45], with fewer cortical sites showing significant changes, although in similar cortical regions. We found power increases in the BHA for both overt and imagined

**Fig. 1 Experimental studies and electrode coverage. a** Study 1 (top row): after a baseline (0.5 s, gray), participants listened to one of six individual words (1 s, light green). A visual cue then appeared on the screen, during which participants were asked to imagine hearing again the same word (1 s, red). Then, a second visual cue appeared, during which participants were asked to repeat the same word (1.5 s, orange). Study 2 (middle row): after a baseline (0.5 s, gray), participants read one of twelve words (2 s, blue). Participants were then asked to imagine saying (red) or to say out loud (orange) this word following the rhythm triggered by two rhythmic auditory cues (dark green). Finally, they pressed a button, still following the rhythm, to conclude the trial. Study 3 (bottom row): after a baseline (0.5 s, gray), participants listened to three auditory repetitions of the same syllable (light green) at different rhythms, after which they were asked to imagine saying (red) or to say out loud the syllable (orange). **b** ECoG electrode coverage across all participants. Yellow, blue and red electrode colors correspond to the study 1, 2, and 3 respectively. **c** Example of one overt and one imagined trial for patient #8 (Geneva study). Time series extracted in the four frequency bands of interest are shown. Vertical lines indicate relevant time events in the study: baseline period (between black lines), word appear on screen (green), auditory cues (red), expected speech time (light blue), actual speech time (dark blue), analysis window (light dashed blue), and manual response (purple).

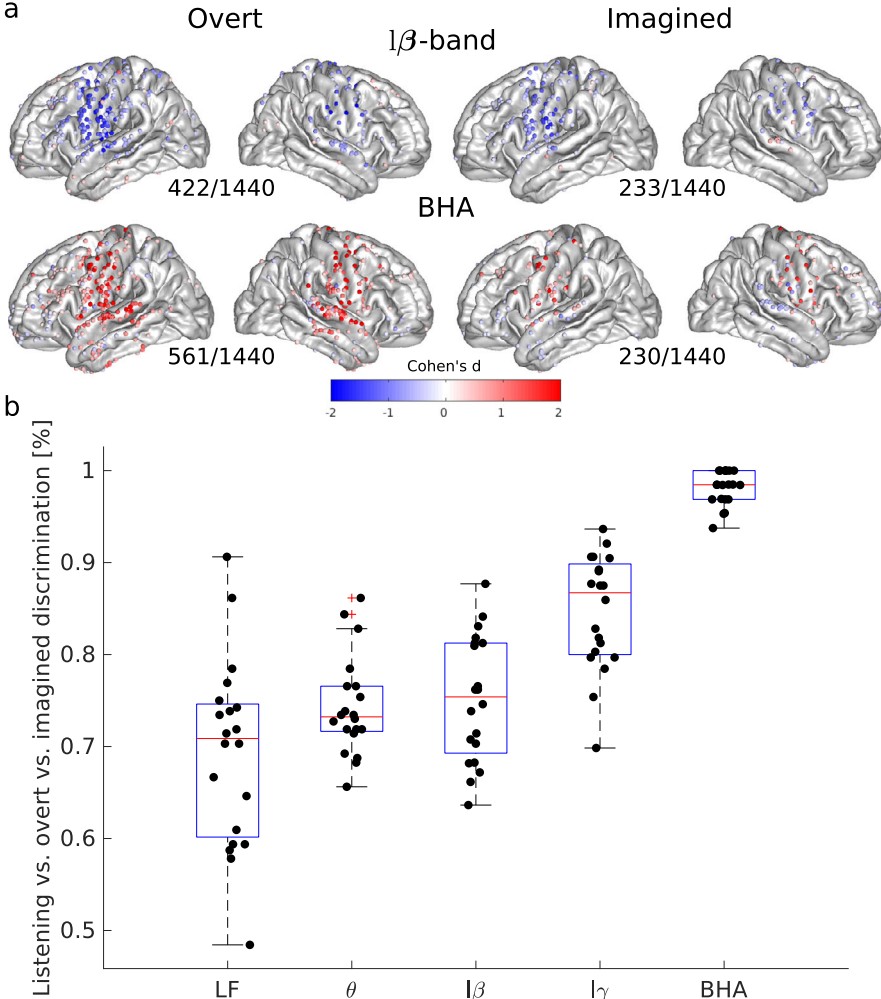

**Fig. 2 Spatial organization and task discriminability of power spectrum deviations from baseline elicited by overt and imagined speech. a** Effect sizes (Cohen's d) for significant cortical sites across all participants and studies during overt and imagined speech compared to baseline (only significant electrodes are shown, t-tests, FDR-corrected, target threshold $\alpha = 0.05$). The number of significant electrodes over the total number of electrodes is indicated below each plot. Left column: overt speech. Right column: imagined speech. Results are pooled across all studies, results for separated studies as shown in Supplementary Figs. 2–4. **b** Discrimination among tasks (listening vs overt vs imagined) using classification with power features for study 1. Features correspond to concatenated time series for power in the different frequency bands. Low-frequency features correspond to ECoG signal amplitude filtered below 20 Hz using a zero-phase Butterworth filter of order 8, rather than power. A linear discriminant analysis classifier was trained using 5-fold cross-validation ($N = 20$). Boxplots' center, bound of box, and whiskers show respectively the median, interquartile range, and the extent of the distribution. Source data are provided as a Source Data file. LF low frequency, lβ low beta, lγ low gamma, BHA broadband high-frequency activity.

speech in the sensory and motor regions (Fig. 2a), and power decrease in the beta band over the same regions. A smaller power decrease was also found over the same regions for the theta and low-gamma bands (Supplementary Fig. 1). Results were consistent whether the participants were imagining speaking or

hearing (Supplementary Figs. 2–4). The most pronounced difference between overt and imagined spatial patterns was that BHA in the superior temporal cortex increased during overt speech, but decreased during imagined speech, presumably reflecting the absence of auditory feedback in the imagined

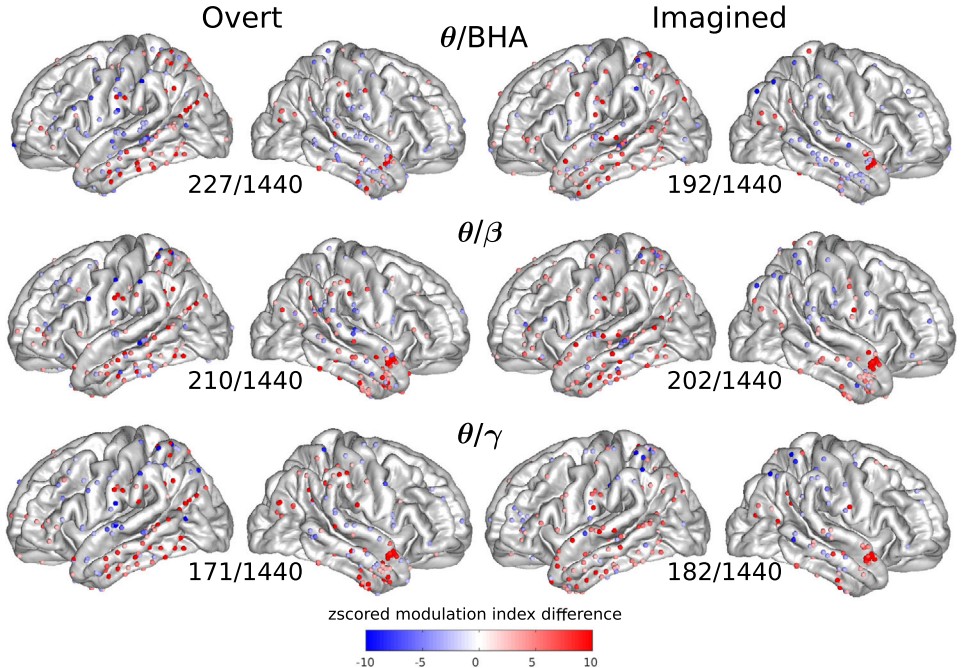

**Fig. 3 Cross-frequency coupling (CFC) between the phase of one frequency band and the amplitude of another (higher) frequency band for each electrode.** Z-scored modulation index difference for significant electrodes across all participants and studies during overt and imagined speech with respect to baseline (only significant electrodes are shown, permutation tests, FDR-corrected, target threshold $\alpha = 0.05$). The number of significant electrodes over the total number of electrodes is indicated below each plot. Left column: overt speech. Right column: imagined speech. Results are pooled across all studies, results for separated studies as shown in Supplementary Figs. 6–8. Source data are provided as a Source Data file. BHA broadband high-frequency activity.

speech situation. The power spectrum differences between overt and imagined speech were sufficiently reliable to accurately classify which task the participants were engaged in. BHA was the most successful feature in discriminating overt and imagined speech, as it was largely absent in the latter (Fig. 2b).

We then quantified phase-amplitude CFC for each cortical site for overt and imagined speech, using the difference in modulation index between speech and baseline periods, taking theta, low-beta, and low-gamma as modulating signal, and beta ($\beta$: 12–25 Hz), gamma ($\gamma$: 25–50 Hz), and BHA as carrier (modulated) signal. This difference was expressed as a z-score relative to its distribution under the null hypothesis, generated with surrogate data using permutation testing. The spatial pattern of cortical sites displaying significant CFC was more widespread than that of power changes. Notably, strong phase-amplitude CFC was found in the left inferior and right anterior temporal lobe between theta phase and other band amplitudes, both for overt and imagined speech (Fig. 3, see Supplementary Fig. 5 for other bands, and Supplementary Figs. 6–8 for each study separately).

Next, we asked whether power spectrum and phase-amplitude CFC variations (hereafter called features) contained information that could be used to discriminate between individual speech words (or syllables in the case of study 3, referred to below as speech items). We systematically quantified the correlation between power spectrum features for all pairs of speech items and their corresponding labels, for each cortical site. We then averaged the resulting correlation across item pairs. As expected, the BHA showed high correlation values for overt speech, primarily within the sensory-motor and superior temporal cortices of both hemispheres, as well as in the left anterior temporal lobe (432/1440 significant electrodes, median of significant values 0.24 [IQR 0.20–0.29], Fig. 4, Supplementary Figs. 9–11 for each study separately). The theta band also showed significant correlations for overt speech in the sensory-motor and

superior temporal cortex (257/1440 significant electrodes, median of significant values 0.22 [IQR 0.19–0.25]). For imagined speech, however, correlations were more diffuse, in particular for the BHA (252/1440 significant electrodes, median of significant values 0.23 [IQR 0.20–0.25]); in the left ventral sensory-motor cortex and bilateral superior temporal cortex, they were lower than for overt speech. Correlations were also observed for the low-beta band in the bilateral superior temporal cortex (287/1440 significant electrodes, median of significant values 0.23 [IQR 0.20–0.24]), and for the theta band in the right temporal cortex only (248/1440 significant electrodes, median of significant values 0.23 [IQR 0.20–0.24]). The same analysis, repeated using phase-amplitude CFC as a discriminant feature, only showed modest correlation values for imagined speech (see Supplementary Figs. 12–15 for the number of significant electrodes in each case).

**Distinct articulatory, phonetic, and vocalic organization between overt and imagined speech.** Based on these initial results, we concluded that the dynamics and neural organization differed for overt and imagined speech production. We, therefore, asked whether the various spatio-temporal organizations of neural activity during overt speech, i.e. the articulatory organization in ventral sensory-motor cortex[24,46], the phonetic organization in superior temporal cortex[47], and the vocalic organization in sensory-motor and superior temporal cortex were conserved during imagined speech. For this, we quantified how well we could discriminate classes in each speech representation system (i.e. labial, coronal, and dorsal for articulatory representation; fricative, nasal, plosive, and approximant for phonetic representation; and low-back, low-front, high-back, high-front and central for vocalic representation; see Methods section). For each region of interest (sensory and motor, middle and inferior temporal, superior temporal, and inferior frontal cortices), we

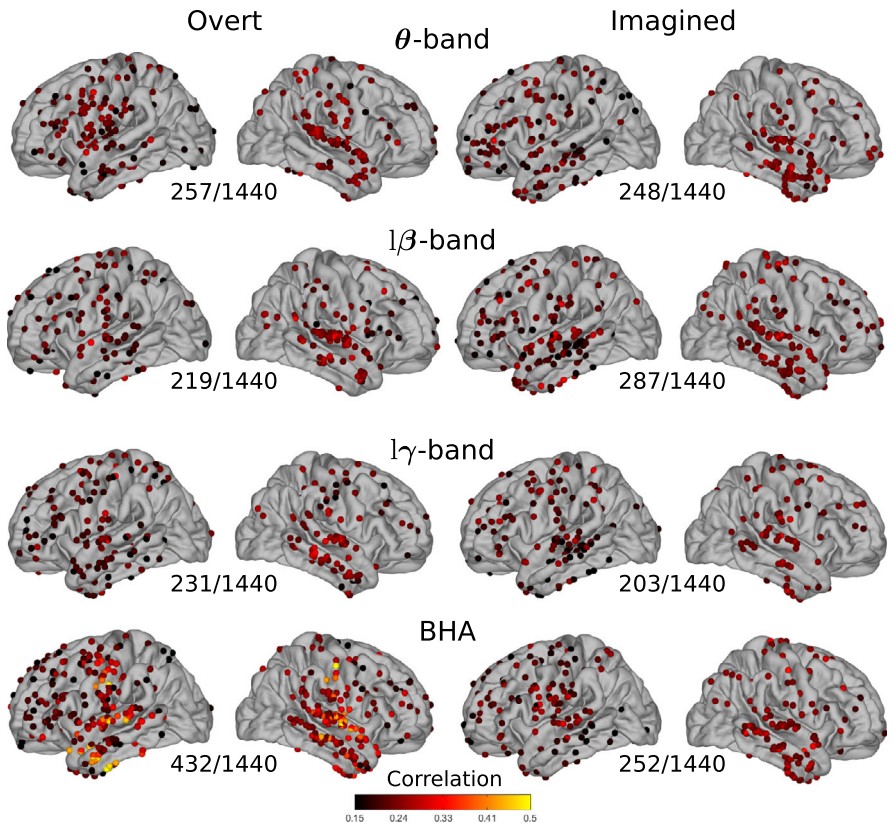

**Fig. 4 Average correlations between individual speech words and their neural representations.** Pairwise correlations between words and power spectrum features averaged across all word pairs for overt and imagined speech on significant electrodes (only significant electrodes are shown, permutation tests, *p* < 0.05, not corrected for multiple comparisons). The number of significant electrodes over the total number of electrodes is indicated below each plot. Left column: overt speech. Right column: imagined speech. Results are pooled across all studies, results for separated studies as shown in Supplementary Figs. 9–11 and 13–15. Source data are provided as a Source Data file. lβ low beta, lγ low gamma, BHA broadband high-frequency activity.

built a high-dimensional feature space in which each basis vector corresponds to one electrode. The dimensionality of this feature space was first reduced with PCA. The Fisher distance (which quantifies features separation) was then computed between each pair of speech items across principal components. As all items were made of one or a sequence of phonemes, and thus belonged to at least one group for each representation, the resulting distance could be attributed to the group(s) that were represented in only one of the two words, i.e. the discriminant one. For instance, the feature distance between the articulatory representations of "python" ([paɪθən], which only includes labial and coronal phonemes) and "cowboys" ([kaʊbɔɪz], which includes a dorsal phoneme in addition to the labial and coronal ones), was assigned to the dorsal group, as it is the dorsal phoneme that is discriminant. In other words, here the dorsal group includes all feature distances for which a dorsal phoneme is present in one word and absent in the other.

For overt speech, as expected, high Fisher distance values were found using the power of the BHA in the sensory-motor cortex (median of significant Fisher distances 0.53 [IQR 0.40–0.72]) and in the temporal lobe (median of significant Fisher distances 0.70 [IQR 0.51–1.19], Fig. 5, see Supplementary Figs. 16 and 17 for each group and study separately). During the imagined speech, however, the BHA was associated with smaller Fisher distances (median of significant Fisher distances 0.40 [IQR 0.32–0.49] across all regions). In fact, lower frequency bands (theta, low-beta, low-gamma) displayed similar or even higher values in left and right hemispheres for phonetic, vocalic, and semantic representations (median of significant Fisher distances 0.45 [IQR 0.30–0.64], 0.40 [IQR 0.35–0.51], and 0.41 [IQR 0.29–0.48] for

theta, low-beta, and low-gamma respectively across all regions). In addition, we disentangled the contribution of each representation to Fisher distance by systematically counting the number of significant Fisher distances across regions for each representation, subject, and band (multi-way ANOVA, Supplementary Table 1; due to collinearity, phonetic and vocalic representations were merged in a single perceptual representation).

Unlike for power spectrum, the Fisher distances for phase-amplitude CFC lay in the same range for overt (median of all significant values 0.38 [IQR 0.26–0.51]) and imagined speech (median of all significant values 0.42 [IQR 0.33–0.65]). Interestingly, the distances were higher in the covert than overt condition in several regions. In the overt speech condition, the highest values were observed for low-beta/gamma phase-amplitude CFC in left sensory-motor and inferior frontal cortices (median of significant values 0.47 [IQR 0.38–0.60]), as well as low-beta/BHA in the left superior temporal lobe (median of significant values 0.44 [IQR 0.39–0.50], Fig. 6, see Supplementary Figs. 18 and 19 for each group and study separately). During the imagined speech, high Fisher distances were obtained mainly for low-beta/BHA phase-amplitude CFC in left sensory-motor cortex (median of significant values 0.80 [IQR 0.72–0.88]) and the right temporal lobe (median of significant values 0.59 [IQR 0.52–0.74]). This was confirmed by performing systematic multi-way ANOVA (Supplementary Table 2).

**Decoding imagined speech.** Finally, we compared the discriminability potential of power spectrum versus phase-amplitude CFC for decoding overt and imagined speech (Fig. 7, Supplementary Fig. 20

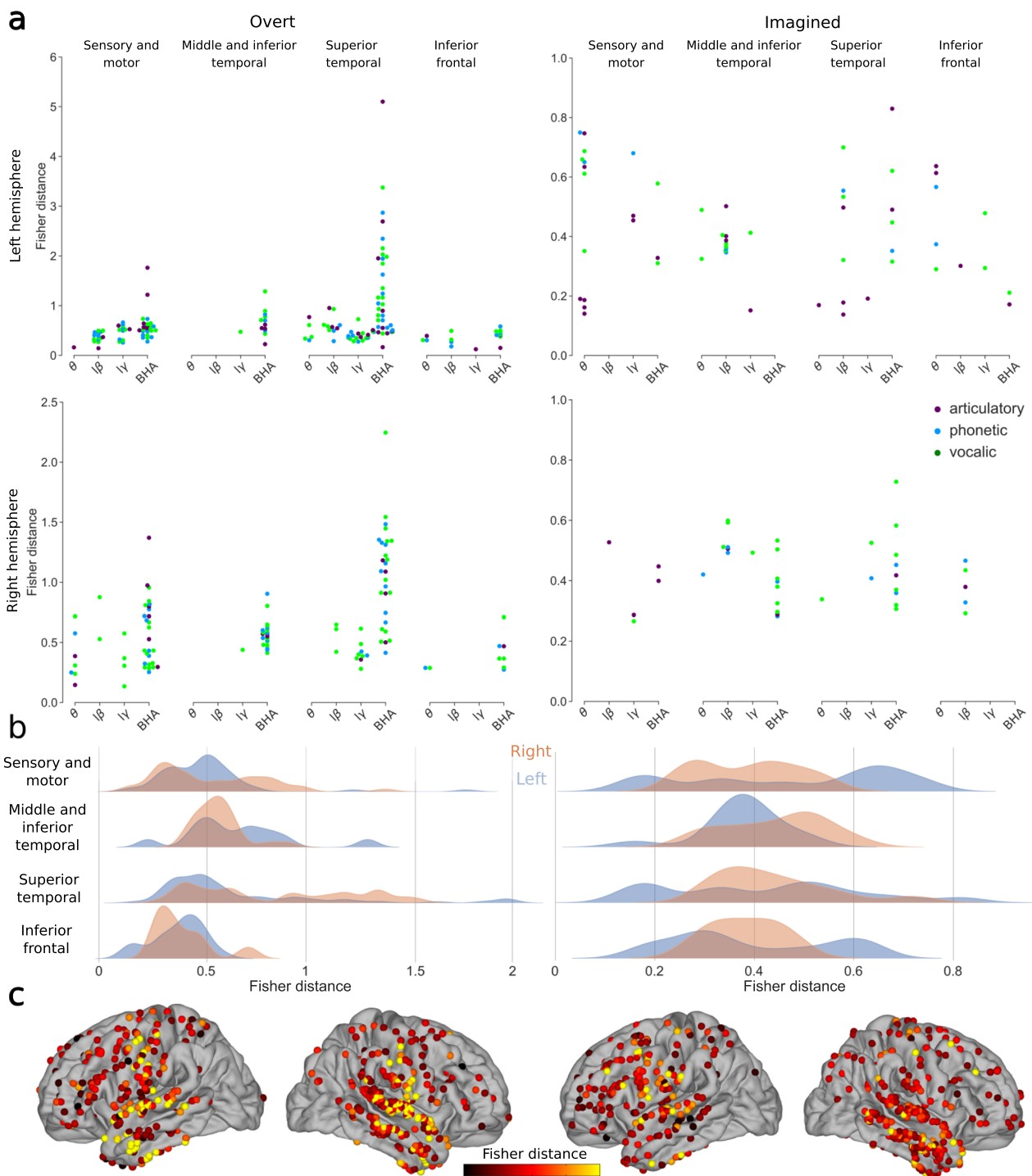

**Fig. 5 Discriminability between different representations using power spectrum for overt and imagined speech. a** Significant Fisher distances between articulatory (purple), phonetic (blue) and vocalic (green) representations in different brain regions and frequency bands (only significant Fisher distances are shown, permutation tests, FDR-corrected, target threshold $\alpha = 0.05$). Note the different scales between overt and imagined speech. **b** Distributions of significant Fisher distances for each brain region and left (blue) and right (orange) hemispheres across all representations and frequency bands (only significant Fisher distances are used, permutation tests, FDR-corrected, target threshold $\alpha = 0.05$). **c** Maximum significant Fisher distance for each electrode across all representations and frequency bands. When several significant Fisher distances were found for the same electrode, only the maximum value is shown and only significant electrodes are shown (permutation test, $p < 0.05$, no FDR correction). Left column: overt speech. Right column: imagined speech. Results are pooled across all studies, results for separated studies as shown in Supplementary Fig. 17. Source data are provided as a Source Data file. lβ low beta, lγ low gamma, BHA broadband high-frequency activity.

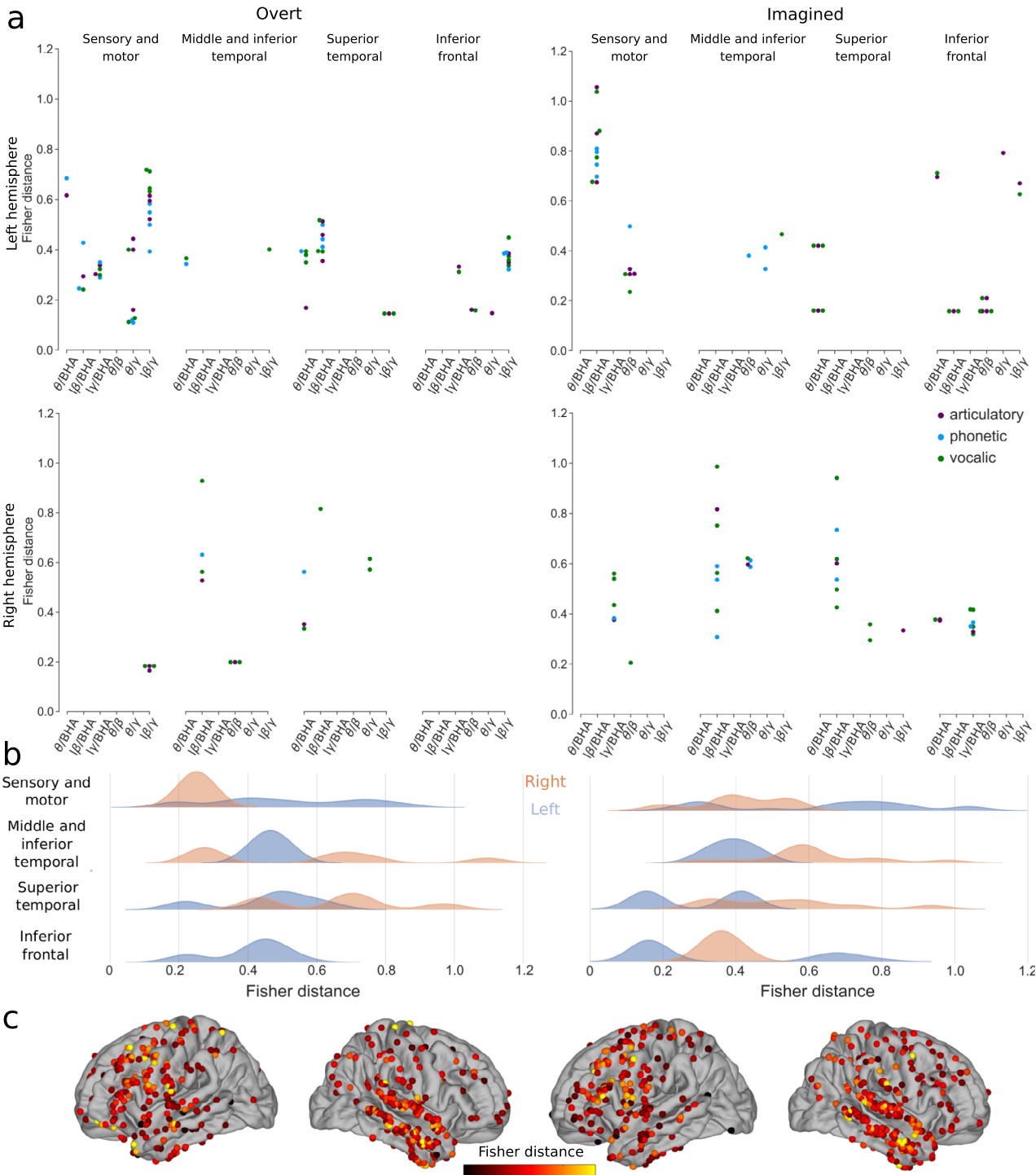

**Fig. 6 Discriminability between different representations using phase-amplitude CFC changes for overt and imagined speech. a** Significant Fisher distances between articulatory (purple), phonetic (blue), and vocalic (green) representations in different brain regions and frequency bands (permutation tests, FDR-corrected, target threshold $\alpha = 0.05$). **b** Distributions of significant Fisher distance for each brain region and left (blue) and right (orange) hemispheres across all representations and frequency bands (permutation tests, FDR-corrected, target threshold $\alpha = 0.05$). **c** Maximum significant Fisher distance for each electrode across all representations and frequency bands. When several significant Fisher distances exist for the same electrode, the maximum value is shown. Only significant electrodes are shown (permutation test, $p < 0.05$, no FDR correction). Left column: overt speech. Right column: imagined speech. Results are pooled across all studies, results for separated studies as shown in Supplementary Fig. 19. Source data are provided as a Source Data file. lβ low beta, lγ low gamma, BHA broadband high-frequency activity.

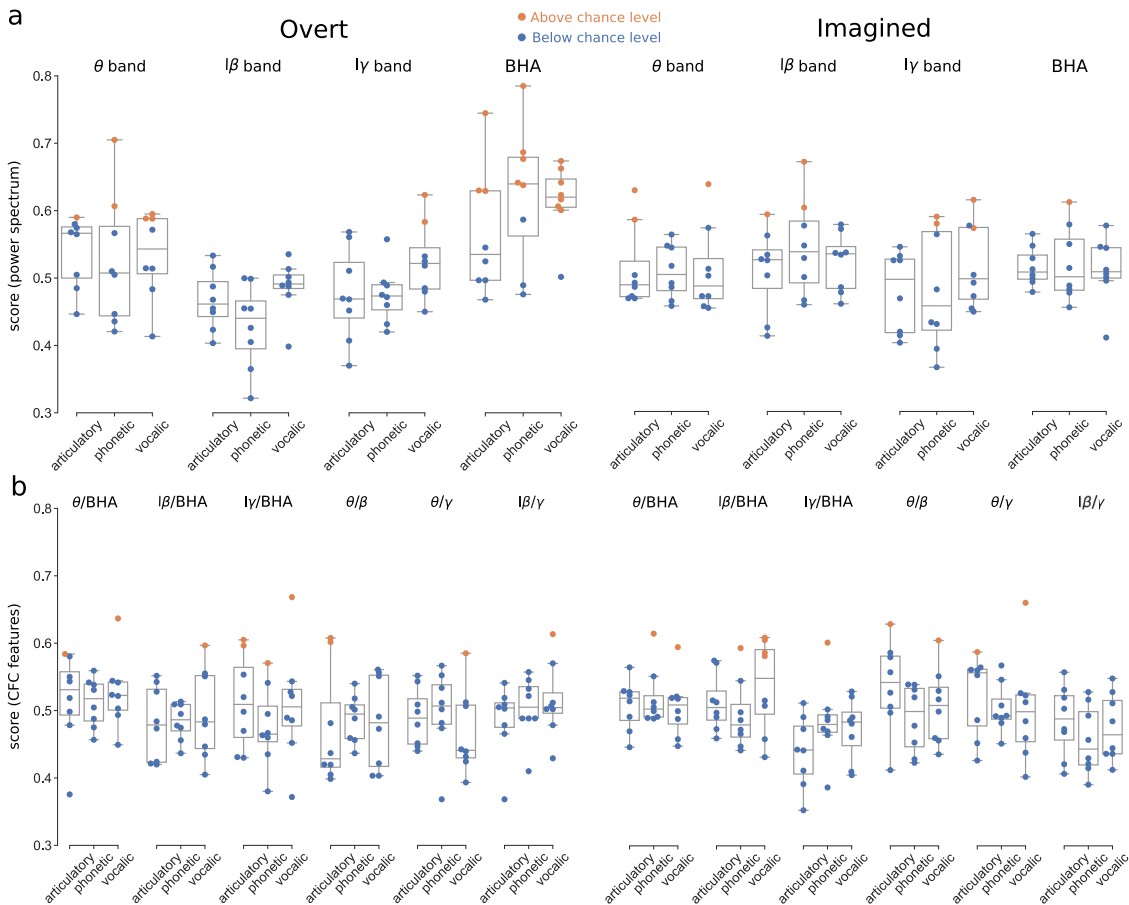

**Fig. 7 Decoding overt (left) and imagined (right) speech.** Orange circles indicate above, significant (vs. blue circles below) chance level performance for each participant respectively ($N = 8$, studies 1 and 2). Boxplots' center, bound of box, and whiskers show respectively the median, interquartile range, and the extent of the distribution (outliers excepted). Significant levels were obtained for each subject based on the number of trials performed (see Methods section). **a** Decoding performance using power spectrum features. **b** Decoding performance using phase-amplitude CFC features. Left column: overt speech. Right column: imagined speech. Source data are provided as a Source Data file. lβ low beta, lγ low gamma, BHA broadband high-frequency activity.

for each study separately). To simplify the decoding problem and to retain enough trials in each class, we grouped the speech items together resulting in a binary classification (study 3 was excluded, as it contained only three syllables). New classes were selected by hierarchical clustering of distances between words according to the articulatory, phonetic, and vocalic representations described above (see Methods section).

For overt speech, significant performance was obtained in 15 participant-representation pairs (median 0.64 [IQR 0.63–0.68]) using BHA power only, and overall, this frequency band worked better than the others (8 significant participant-representation pairs, median 0.59 [IQR 0.59–0.61]). For imagined speech, however, decoding based on BHA power (1 significant participant-representation pair, performance 0.61) did not yield better scores than with other bands (10 significant participant-representation pairs, median performance 0.60 [IQR 0.59–0.63]) which were equally good using e.g., theta or beta power. We also observed that decoding worked better for phonetic and vocalic (i.e. perceptual) representations than for the articulatory one, which supports the flexible abstraction hypothesis of imagined speech, i.e. that it is defined at the phonemic rather than the motor level, with some degree of flexibility across experiments and individuals[26,31–33]. Importantly, the decoding performance for overt speech increased significantly when the trials were realigned using the participant's speech output (vocal onset), suggesting that imagined performance would improve as well if a consistent way of realigning trials could be found (Supplementary Fig. 21).

Interestingly, when we used phase-amplitude CFC as a feature, decoding did not perform better for overt (11 participant-representation pairs above chance level, median 0.60 [IQR 0.59–0.61]) than imagined speech (12 participant-representation pairs above chance level, median 0.60 [IQR 0.59–0.61]). No participant was consistently better than the others across frequency bands and representations. No specific frequency band stood out for overt speech, although the articulatory and vocalic representation worked better. For imagined speech, the low-beta/ BHA performed better than other phase-amplitude CFC for imagined speech, confirming the results found in Fig. 6, particularly for the perceptual representations.

## Discussion

We examined the neural processes underlying the production of overt and imagined speech, in order to identify features that could be used for decoding imagined speech, having in sight a potential future application to severe speech production deficits. Importantly, we assessed whether the features that are most easily decodable for imagined speech are similar or different from those that work best for overt speech. To do so, we did not only explore the articulatory dimension, but also perceptual (phonetic and vocalic) representation spaces. We found that overt and imagined speech differed in some crucial aspects of their oscillatory

dynamics and functional neuroanatomy. First, while the articulatory representation was well encoded in overt speech, other representations, especially the perceptual one, better reflected imagined speech. Overt and imagined speech both engaged a large part of the left hemispheric language network, with a more prominent involvement of the superior temporal gyrus for overt speech, presumably because of auditory feedback processes. Second, while BHA showed the best performance for overt speech decoding, it conveyed little word- or syllable-specific information during imagined speech. Conversely, neural activity at lower frequencies could be used to decode imagined speech with equivalent or even higher performance than overt speech.

These results suggest that it might prove difficult to successfully transfer the decoding process of brain-computer interfaces trained with overt or even silently articulated speech to imagined speech[2,4,5]. BHA representations were poorly discriminant in primary sensory and motor regions during the imagined speech, in accord with the flexible abstraction hypothesis of imagined speech, suggesting that BHA might not be a good feature for imagined speech decoding. We also found that the beta-band stood out as a prominent feature in the neural encoding of imagined speech, both in terms of power and CFC (low-beta/gamma and low-beta/BHA), with slightly better performance when using power. This finding aligns well with the notion that the beta band plays an important role in endogenous processes, notably in relation with top-down control in the language domain[16,17,40,43,48,49]. Although repeating a heard or written word engages automatic, almost reflex, neural routines, imagined speech is a more voluntary action requiring enhanced endogenous control from action planning frontal regions[50–52]. Since spurious CFC can sometimes result from non-linearity, non-stationarity, and power changes across conditions in the signal[53,54], we carefully selected spectral peaks for the modulating signal to ensure a well-defined phase and specific bandwidths for the modulated signal. Despite this precaution, we would not in principle exclude that significant CFC coupling reflects other types of changes from baseline to signal. Yet, that significant and specific decoding performance could be obtained with these features suggests that they do contain information that is specific to individual speech items.

Decoding performance for overt speech increased significantly when trials were aligned on recorded speech onset (>80% for 3 over 4 participants for a 6-class decoding task, Supplementary Fig. 21), which are obviously absent for imagined speech. This good performance is expected as high decoding accuracy can be reached on intracranial signals associated with overtly pronounced sentences[2–4]. Previous attempts to align imagined speech directly based on neural data[25] met limited success due to the large variability of neural signals across trials, the low signal-to-noise ratio, and possibly also the jitter in covert speech onsets across trials. Future work will be necessary to test alternative methods based on an extension of these ideas[55,56]. Although decoding performance would presumably increase if imagined speech onsets and offsets were more reliably detected, we show here that imagined speech decoding is possible using features, such as phase-amplitude CFC, that do not require precise alignment of single-trial data. The absence of behavioral output during imagined speech might even be an advantage, as it definitely prevents the contamination of neural signal recordings by the participant's voice, a problem that was recently raised. Because the fundamental frequency of the human voice overlaps with the neural BHA, an acousto-electric effect might have artificially inflated the performance in previous overt speech decoding studies[57]. To enable a fair comparison of overt and imagined speech in our study, we checked that the three current datasets were free of acoustic contamination (Supplementary Fig. 22). A

further technical advantage of silent speech is the absence of movement artefacts. In the three presented studies, the task instructions explicitly stated that participants should not articulate. On-line and off-line (audio/video recordings) monitoring of the patients, confirmed that they did not silently mouth or whisper words (see Methods section), even though we could not control for some degree of silent mouthing. Note that although we could have expected less residual motion after the instruction "imagine hearing" (study 1) than "imagine speaking" (studies 2 and 3), our results did not confirm any such bias (Supplementary Fig. 20).

The originality of the study is that we explored a broad feature space and a large brain coverage to answer the important questions of which brain regions and features should be targeted for a cortical speech BCI. We find the new frequency and cross-frequency features allowing for decoding with >60% accuracy (in a binary SVM classification setting). Previous studies attempting to decode covert speech using only BHA features obtained performances similar to the performance we obtained with BHA features[18,19] (even though the quantitative comparison is difficult considering differences in study designs), which supports the interest in broadening the spectrum of features that can potentially be useful. Although the signal-to-noise-ratio remains low in our setting, and binary classification far from a real imagined speech BCI setting, and considering that we only have access to a limited number of non-aligned trials in the epilepsy model, we can expect far better results in real conditions with a patient chronically implanted on purpose. In that case, working with all the presently identified features, as well as sampling from temporal/auditory regions will greatly enhance the chance of clinical success. Despite these obvious limitations, the current results allow us to formulate a number of concrete proposals for the design of future speech BCIs. In this study, we could probe the representations of imagined speech at various linguistic levels, namely articulatory, phonological, and vocalic. Notwithstanding the typical weakness of imagined speech signals, we reached significant decoding performance using lower frequencies and the phonetic representation level. While this is good news for future speech BCIs, the word level, which was mostly used in this study, is presumably not the optimal currency for an efficient imagined speech decoding strategy based on phonetic representations. A realistic BCI will have to offer decoding based on representation space that can span the size of the average human language repertoire. Likewise, while we showed potential separation in the feature space of syllables, a phoneme decoding strategy would suffer from the combinatorial explosion issue. Using a restricted set of the most commonly used syllables, including monosyllabic morphemes, which patients could combine to convey their basic needs, might be an interesting first approach. Such a strategy would presumably benefit from the syllable feature space separation shown here. In the future, introducing even more complex, sentence-level stimuli, rather than single words or syllables, could further allow exploiting additional representation levels for imagined speech decoding, such as inference, long-term memory, prosody, semantic mapping, etc.[58–60], bringing us closer to ecological and generalizable conditions[61,62]. This will also maximize the number of available trials and allow implementing more complex machine learning such as deep learning. Each presented stimulus triggers neural activity that might be influenced by word length, frequency, emotional valence, in addition to syntactic and semantic content[10,63]. The richness of these contextual cues could turn out to be an advantage, as it could maximize the separability of speech items, leading to easier decoding, regardless of the representation. In future imagined speech decoding BCIs, this will also enable the exploration of other questions about imagined speech, such as whether the

prosodic patterns of overt speech are conserved, or whether the neural representations of the feature space remain consistent across subjects.

Our results build upon three studies with distinct instructions for the participant (imagine hearing for study 1, imaging speaking for study 2 and 3), allowing for probing whether imagine hearing and imagine speaking are underpinned by the same or distinct brain networks. While some similarities arise from the results of studies 1 and 2 (Supplementary Figs. 17 and 19), the poor overlap in the electrode coverages between the two studies does not allow for a strong conclusion. The cerebral substrates of imagining oneself speak vs. imagining hearing speech are known to overlap significantly, but there are also important differences. Imagined speaking recruits fronto-parietal sensorimotor regions more strongly, suggesting that this task involves a motor-to-sensory transformation. On the contrary, imagined hearing activates more the inferior parietal cortex and intraparietal sulcus, which could correspond to a sensory memory-retrieval operation[64,65]. Of note, most studies on the topic did not specifically contrast imagining oneself speak vs. imagining hearing one's own voice[66]. In a single-subject fMRI study[67], the participant activated her left inferior frontal gyrus more when she reported imagining speaking vs. hearing herself speak. Further studies with large electrode coverage and testing these two task instructions will be essential as an important alternative for patients with a certain type of post-stroke aphasia, e.g. Broca. The optimal cue (e.g. hearing vs. reading) will also need to be investigated as a potential source of improvement, although the present data do not allow us to conclude on this point.

Adding more features, and especially higher-level ones could also help addressing the ethical question of which part exactly of the imagined speech should we let machines decode[68]. Finally, decoding performance will likely dramatically increase when using on-line systems where both the subject and the algorithm learn simultaneously, which so far is difficult to envisage in patients who are implanted over a short time span as in epilepsy.

Our results highlight the large variability in the best decoding features across participants and tasks for imagined speech, suggesting that decoding strategies, i.e. a specific set of spatial and frequency features (anatomical regions, frequency bands, and specific tasks) will have to be adjusted individually in order to build efficient imagined speech BCI systems. In that respect, low frequencies might be more powerful features to decode from spatio-temporally variable signals than BHA, since they tend to be both spatially coherent over larger areas of the cortex and temporally less constrained[16]. By indexing a more integrated neural activity, they might distinguish better the different imagined speech items. Even though they presumably carry less content-specific information than BHA, they seem to be a necessary complement to distributed focal high-frequency activity. In addition, lower frequency bands, such as delta, are expected to become prominent features in more natural BCI setting where patients will attempt to produce longer speech sequences[38].

The interest of taking into account lower frequency bands has practical consequences for the design and placement of intracranial electrodes. Imagined speech decoding based on both local (BHA) and large-scale signals will become possible in the near future using a new generation of soft, ultra-flexible, high-density ECoG electrodes that could maximize brain coverage, as well as the amount and quality of the contacts with the cortex. Active multiplexing and graphene (nanomaterial)-based neural interfaces are two areas of active research in the field[69]. With such electrodes and related wireless electronics, on-line signal analysis will be easier, for a more convenient use with BCIs. Off-line analyses such as those we present here are the first and necessary step to guide us once we will be able to use those electrodes in humans along with on-line systems. Unlike the robotic arms that are currently being developed for motor restoration, which are optimally controlled by the dense sampling of a spatially restricted cortical area (typically a Utah array)[1,7], a language BCI system for severe aphasia will require broader coverage of the cortical surface, including the frontal and the temporal lobes, to not only cope with the high physiological intersubject variability of inner speech production but also with the variable structural damage (cortical, subcortical) that patients may have suffered from. In post-stroke Broca-type aphasia, the efforts to overcome the overt speech planning deficit during imagined speech are expected to implicate a large range of regions of the language network, which will all have to be sampled.

## Methods

**Participants**. Electrocorticographic (ECoG) recordings were obtained in 3 distinct studies from 13 patients (study 1: 4 participants, 4 women, mean age 25.6 years, range 19–33; study 2: 4 participants, 3 women, mean age 30.5 years, range 20–49; study 3: 5 participants, 3 women, mean age 32.6 years, range 23–42) with refractory epilepsy using subdural electrode arrays implanted as part of the standard pre-surgical evaluation process (Supplementary Table 3). Electrode array locations were thus based solely on the requirements of the clinical evaluation. Participants were recruited from three medical centers: Albany Medical Center (NY, USA), Geneva University Hospitals (Switzerland), and NYU Langone Medical Center (NY, USA). All participants gave informed consent, and the experiments reported here were approved by the respective ethical committees (Albany Medical College Institutional Review Board[18], Commission Cantonale d'Ethique de la Recherche, project number 2016-01856, and the Institutional Review Board at the New York University Langone Medical Center). No monetary compensation was given to the participants.

**Studies and data acquisition**. Three distinct experiments were performed, one in each study center. In all three studies, participants were closely monitored by the experimenter to prevent any residual movements. The absence of motion was further checked post-hoc by visual inspection of video recordings. Participants also confirmed after the experiment that they followed instructions (i.e. using imagine hearing or imagine speaking strategy).

*Study 1: word repetition*. The first study was a word repetition paradigm (Fig. 1a). This data appeared first in[18]. The participant first heard one of six words presented through a loudspeaker (average length: 800 ms ± 20). A first cross was then displayed on the screen (1500 ms after trial onset) for 1000 ms, indicating that the participant had to imagine hearing the word. Finally, a second cross was displayed on the screen (3000 ms after trial onset) for a duration of 1500 ms, indicating that the participant had to repeat out loud the word. The six words ('spoon','cowboys','battlefield','swimming','python','telephone') were chosen to maximize the variability of acoustic representations, semantic categories, and the number of syllables while minimizing the variability of acoustic duration. Participants performed from 18 to 24 trials for each word.

Implanted ECoG grids (Ad-Tech Medical Corp., Racine, WI; PMT Corporation, Chanhassen, MN) were platinum-iridium electrodes (4 mm in diameter, 2.3 mm exposed) embedded in silicon. Inter-electrode distance was 4 or 10 mm. ECoG signals were recorded using seven 16-channel g.USBamp biosignal acquisition devices (g.tex, Graz, Austria) with a 9600 Hz sampling rate. Reference and ground were chosen by selecting ECoG contacts away from epileptic foci and regions of interest. Data acquisition and synchronization with task stimuli were performed with the BCI2000 software[70]. The participant's voice was also acquired through a dynamic microphone (Samson R21s) that was rated for voice recordings (bandwidth 80-12000 Hz, sensitivity 2.24 mV/Pa) placed 10 cm away from the patient's face. Another dedicated 16-channel g.USBamp amplifier was used to acquire and digitize the microphone signal to guarantee synchronization with ECoG data. Finally, the participants' compliance with the imagined task was verified with an eye-tracker (Tobii T60, Tobii Sweden).

*Study 2: rhythmic word repetition*. The second study was also a word repetition paradigm (Fig. 1b). The participant first read one of twelve words presented on a laptop screen for 2000 ms. Two successive auditory cues were then presented through a loudspeaker (2100 ms and 2900 ms after the beginning of the trial). The participant then had to repeat out loud or imagine saying the word following the rhythm given by the two auditory cues (i.e. participant output was expected to start at around 3700 ms). Finally, following the same rhythm, the participant would press a key on the laptop's keyboard (expected at around 4500 ms). Participants were repeating French words (for three participants;'pousser','manger','courir','pallier','penser','élire','enfant','lumière','girafe','état','mensonge','bonheur') or similar German words (for one participant;'schieben','essen','laufen','leben','denken','wählen','Kind','Licht','Giraffe','Staat','Treue','Komfort'). Words were

chosen to belong to four different semantic categories (concrete verbs, abstract verbs, concrete nouns, abstract nouns). Participants performed from 7 to 15 trials for each word. Stimuli presentation was performed with Psychotoolbox.

ECoG signals were acquired by subdural electrode grids and strips (Ad-Tech Medical Corp; inter-electrode distance: 4 or 10 mm), amplified and digitized at 2048 Hz and stored for offline analysis (Brain Quick LTM, Micromed, S.p.A., Mogliano Veneto, Italy).

*Study 3: rhythmic syllabic repetition*. The third study was a syllable repetition paradigm (Fig. 1c). A syllable was presented sequentially three times on a loud-speaker. The time interval between repetitions was selected randomly for each trial from one of three possibilities (800 ms, 1000 ms, 1200 ms). Following the same rhythm given by these syllables, the participant then had to repeat out loud or imagine saying the syllable. Participants were repeating one of three syllables ('ba','da','ga') in each trial. These syllables were chosen to minimally differ acoustically (by a few dozens of ms of voice onset time, VOT) but rely on very different movements at the articulatory levels. This aims at optimizing the differences observed at the production level while limiting potential contamination by exogenous acoustic cues. Participants performed 16–55 trials for each syllable. Stimuli presentation was coordinated with Presentation software (Neurobehavioral Systems).

All behavioral recordings were made via a computer on the service tray of a hospital bed using Presentation Software (NeuroBehavioral Systems). Audio recordings were obtained using a microphone connected to the computer, and were synchronized to the onset of the last auditory cue. Electroencephalographic (ECoG) activity was recorded from intracranially implanted subdural electrodes (AdTech Medical Instrument Corp.). Recordings included grid, depth and strip electrode arrays. Each electrode had a diameter of 4 mm (2.3 mm exposure), and the space between electrodes was 6 mm (10 mm center to center). Neural signals were recorded on a 128-channel Nicolet One EEG system with a sampling rate of 512 Hz.

**Anatomical localization of ECoG electrodes**. ECoG electrodes were localized using the iELVis toolbox (http://github.com/iELVis/iELVis)[71]. Briefly, each patient's pre-implant high-resolution structural MRI scan was automatically segmented and parcellated using Freesurfer (http://surfer.nmr.mgh.harvard.edu/)[72]. A post-implantation high-resolution CT or MRI scan was coregistered with the pre-implant MRI scan. Electrode artifacts were identified visually on the postimplant scan. Electrode coordinates were corrected for the brain shift caused by the implantation procedure by projecting them back to the pre-implant leptomeningeal surface. Electrode coordinates from individual participants were brought onto a common template for plotting.

**Signal processing**. To confirm that participants did not silently mouth or whisper words, three controls were performed: (i) During the experiment, the experimenter was checking visually and by hearing that the participant was not whispering or silently mouthing words, and correcting him/her if necessary. (ii) Recorded audio of each experiment and patient were listened to by two of the authors to check that no speech could be heard during the imagined speech segment. One patient from the first study was not included in the present dataset based on this control, as the patient was whispering words instead of pronouncing them covertly. (iii) Available video recordings of patients during the task were watched by one of the authors to check that no movements of the mouth could be observed during the imagined speech task.

Time series were visually inspected, and contacts or trials containing epileptic activity and excessive noise were removed. Trials with overt speech were checked for acoustic contamination by correlating the recorded audio signal and the neural data (Supplementary Fig. 22)[57]. All times series were then corrected for DC shifts by using a high-pass filter with a 0.5 Hz cutoff frequency (zero-phase Butterworth filter of order 6, zero-pole-gain design). Electromagnetic noise was removed using notch filters (forward-backward Butterworth filter of order 6, zero-pole-gain design, cutoff frequencies: 58–62 Hz, 118–122 Hz, and 178–182 Hz for studies 1 and 3; 48–52 Hz, 98–102 Hz, 148–152 Hz, and 198–202 Hz for study 2). Finally, times series were re-referenced to a common average, and down-sampled to a new sampling rate of 400 Hz, 400 Hz, and 512 Hz for studies 1, 2, and 3 respectively using a finite impulse response antialiasing low-pass filter. Periods of interest for imagined and overt speech were selected either during the period with a visual cue (study 1), or 250 ms before to 250 ms after the expected production time (studies 2 and 3).

**Power spectrum**. Time series were transformed to the spectral domain using an analytic Morlet wavelet transform. The power spectrum was then obtained by taking, for each frequency band, the averaged (over frequencies and time epochs of interest) absolute value of the complex spectral time series. We did not normalize each band independently before averaging, as normalizing only induced very limited changes in the resulting powers of each band compared to when no normalization was applied. The four frequency bands of interest were the theta (θ, 4–8 Hz), low-beta (lβ, 12–18 Hz), and low-gamma (lγ, 25–35 Hz) bands, and the

broadband high-frequency activity (BHA, 80–150 Hz). Cohen's effect size $d = (\bar{x}_1 - \bar{x}_2)/s$ was assessed by computing the difference between the mean of the power spectrum distribution for all trials during overt or imagined speech and the mean of the power spectrum distribution during baseline for all corresponding trials, divided by the pooled standard deviation $s = \sqrt{((n_1 - 1)s_1^2 + (n_2 - 1)s_2^2)/(n_1 + n_2 - 2)}$, with $n_i$ and $s_i$ respectively the number of samples and the variance in distributions $i \in \{1, 2\}$. Significance was assessed by rejecting the null hypothesis of equality of the mean of both distributions with a two-tailed, two-sample t-test, corrected for multiple comparisons using the Benjamini–Hochberg false discovery rate (FDR) procedure (target $\alpha = 0.05$)[73].

**Phase-amplitude cross-frequency coupling**. Phase-amplitude cross-frequency coupling (CFC) was assessed between the phase of one band and the amplitude of a higher-frequency band[74]. To ensure that the phase of the modulating (lower) band was well defined[53], we first identified peaks in the log power spectrum for each electrode. Then, for each modulating frequency band of interest (theta band: θ, 4–8 Hz, low-beta band: lβ, 12–18 Hz, and low-gamma band: lγ, 25–35 Hz), the peak with maximal amplitude, if existing, was selected. The modulating band was then obtained by filtering original data for each modulating frequency band with a band-pass filter centered around each peak frequency with a bandwidth equal to half the size of the band of interest (i.e. 2 Hz, 3 Hz, and 5 Hz for a peak in the theta, low-beta, or low-gamma band respectively). To ensure that the modulated (higher) band was large enough to contain the side peaks produced by the modulating band, we increased the bandwidth when necessary for the modulated frequency of interest (beta band: β, 12–25 Hz, gamma band: γ, 25–50 Hz, broadband high-gamma activity: BHA, 80–150 Hz)[53]. Despite these precautions, we cannot expect that the theta/beta and low-beta/gamma phase-amplitude CFCs to be fully represented due to the limited bandwidth we can afford for the modulated frequency. The band-pass filter was a zero-phase Butterworth filter of order 6 with zero-pole-gain design. The phase and amplitude were then obtained using the Hilbert transform of the centered filtered signals.

Then, for each time-epoch of interest, the histogram (18 bins) of amplitudes as a function of phases was computed and averaged across trials. Modulation index (MI) values were then calculated from the Kullback-Leibler divergence (KL) between the averaged histogram of the signal and the uniform distribution as $MI = KL/\log(\#bins)$[74]. Z-scores for MI were computed by comparing the observed difference between MI values of overt/imagined time epochs and baseline $x_d$ with the surrogate distribution of differences between MI values of overt/imagined time epochs and baseline $x_{ds}$ as $z = (x_d - \bar{x}_{ds})/s_{sd}$, with $s_{sd}$ the standard deviation of the surrogate distribution. Surrogates were obtained by randomly shuffling 200 times the overt/imagined time epochs and baseline distribution.

One-tailed p-values corresponding to the z-scores were obtained from the cumulative normal distribution (one-tailed since the observed MI can only be greater than the surrogate one, not smaller), FDR-corrected for multiple comparisons (target $\alpha = 0.05$)[73].

**Pairwise correlation of features with words**. Pairwise correlation was quantified by computing for each speech items the Pearson's correlation between power spectrum or phase-amplitude CFC features and the labels. Labels were set to 1 and −1 for the first and second word or syllable respectively of the pairwise comparison. The average pairwise correlation was then obtained for each electrode by averaging pairwise correlations across all pairs of speech items. Statistical significance was assessed by random permutations: for each speech item pair, labels were randomly permuted, and the procedure was repeated 1000 times. A null distribution was then obtained by averaging across all speech item pairs. Significant values are those for which the p value is <0.05, without correction for the number of electrodes.

**Articulatory, phonetic, and vocalic representations**. Words were decomposed according to their phonetic content by finding articulatory, phonetic, and vocalic groups for each phoneme contained in the word (Supplementary Tables 4 and 5). Each word was thus represented by a set of different groups for each representational dimension. For instance, the word 'python' [paɪθən] was represented as labial ([p]) and coronal ([θ], [n]) for the articulatory representation, plosive ([p]), fricative ([θ]), and nasal ([n]) for the phonetic representation, and low-front ([a]), high-front ([ɪ]), and central for the vocalic one ([ə]). Discriminability (feature distance) between two words was then assigned only to the groups that were present in one of the two words for each representation.

Discriminability to compare two words $i$ and $j$ were computed using the Fisher distance between their power-spectrum or cross-frequency coupling feature distributions. Fisher distance was defined as:

$$\max_{j \in [1..n_j]} = \frac{(\mu_i - \mu_j)^2}{(\sigma_i^2 + \sigma_j^2)} \tag{1}$$

with $\mu_i$ and $\sigma_i$ the mean and standard deviation of the feature distribution respectively, $n_j$ the dimensionality of features. Correlation could have been used as well as another metric of discriminability. The resulting values were then averaged

across instances for each patient and each group. Statistical significance was assessed by random permutations: for each pair of speech items, labels were randomly permuted, and the procedure was repeated 1000 times. A null distribution was then obtained by averaging across each instance for each patient and each group. Significant values values were found after FDR-correction for multiple comparisons (target $\alpha = 0.05$).

**Decoding**. For articulatory, phonetic, and vocalic decoding, word labels were grouped together in two new classes by computing the distance between labels according to each specific representation. Distance between two words was incremented by 1 for each phoneme's group that was present only in one of the two words. Hierarchical clustering was then performed on the resulting distance matrix between all pairs of words (linkage criterion that uses the maximum distances between all observations of the two sets of observations). The new classes were selected by taking word groups that were close-by in the dendrogram, while minimizing the class imbalance.

We trained a specific classifier for each binary classification problem resulting from this clustering procedure. We used a 10-fold cross-validation approach, i.e. data was divided into 10 blocks, with 90% of the blocks being used for training, and the remaining block being used for testing. This procedure was repeated 10 times by shifting every time the block used for testing. We used a support vector machine algorithm with a linear kernel for classification. Due to the low number of available trials, we preferred this robust linear approach to more complex, non-linear machine learning techniques. Feature selection was done using recursive feature elimination, starting with the full set of features and removing sequentially features that do not contribute to the classifier performance. Feature selection was done using nested 5-fold cross-validation within the training set. The score was evaluated using balanced accuracy to account for class imbalance that could occur when there were more samples in one of the two classes.

Thresholds for significant classification performance were obtained independently for each subject from an inverse binomial distribution, which accounts for the possibility of obtaining by chance accuracies higher than 50% in a binary classification problem because of a low number of trials[75].

**Reporting summary**. Further information on research design is available in the Nature Research Reporting Summary linked to this article.

## Data availability

Raw patient-related data are protected and are not available due to data privacy laws. Processed neurophysiological and neuroanatomical data are available under restricted access for ethical and privacy reasons. Access to data collected at Geneva University Hospitals can be requested by contacting Pierre Mégevand (pierre.megevand@unige.ch) and is conditional to the establishment of a specific data sharing agreement between the applicant's institution and the University of Geneva. Source data are provided with this paper.

## Code availability

Code was written in MATLAB and Python, and is available at https://doi.org/10.5281/zenodo.5702872.

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

## Acknowledgements

This work was funded by EU FET-BrainCom project (A.G.), NCCR Evolving Language, Swiss National Science Foundation Agreement #51NF40_180888 (A.G.), NINDS R3723115 (R.T.K.), Swiss National Science Foundation project grant 163040 (A.G.), National Natural Science Foundation of China 32071099 (X.T.), Natural Science Foundation of Shanghai 20ZR1472100 (X.T.), Program of Introducing Talents of Discipline to Universities, Base B16018 (X.T.), NYU Shanghai Boost Fund (X.T.), Fondation Pour l'Audition FPA RD-2020-10 (L.A.), Swiss National Science Foundation career grant 167836 (P.M.), and Swiss National Science Foundation career grant 193542 (T.P.). The authors thanks Dr. Gerwin Schalk, Dr. Dan Friedman, and Dr. Patricia Dugan for providing access to the datasets used in this work, and Dr. Johanna Nicolle for sharing her linguistic expertise.

## Author contributions

S.M., X.T., L.A., P.M., and A.G. designed the experiments. A.C., X.T., and L.A. collected the data. T.P. and J.D. performed the analysis. T.P., L.A., P.M., and A.G. drafted the manuscript. T.P., J.D., A.C., S.M., B.P., R.K., X.T., D.P., W.D., O.D., L.A., P.M., and A.G. corrected and approved the manuscript.

## Competing interests

The authors declare no competing interests.

## Additional information

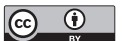

