## [Peer Review File · Nature Communications]

Imagined speech can be decoded from low- and cross-frequency intracranial EEG featuresREVIEWER COMMENTS

Reviewer #1 (Remarks to the Author):

SUMMARY:

This study led by Proix and Saa uses electrocorticography (ECoG) recordings in people undergoing epilepsy monitoring to study the electrophysiological correlates of overt (spoken out loud) and imagined speech. The manuscript's primary focus is to characterize imagined speech (vis-à-vis its differences compared to overt speech), with the motivation that this knowledge will help inform the design of speech brain-computer interfaces (BCIs) to help patients with severe speech production impairments. The authors compare the relationship between word properties (articulatory, phonetic, and vocalic) with a number of electrical field potential features (i.e., four different frequency band powers and cross-frequency coupling between bands), in four different anatomical areas (sensorimotor, middle/inferior temporal, superior temporal, and inferior frontal). They report a number of findings including that although broadband LFP was best correlated with overt speech (consistent with a number of previous studies), for decoding imagined speech it was comparable (and both were quite poor) to lower frequency features and phase-amplitude coupling. These results provide some insight into future directions for trying to improve imaginary speech BCIs, and represent a rare and extensive collection of neural recordings in people during both overt and imaginary speaking tasks.

STUDY STRENGTHS:

The study of imagined speech is timely, as speech BCIs based on decoding overt speech have progressed rapidly in recent years, but so far there has been comparatively much less work in speech BCIs for people who cannot actually speak (and imagined speech is one highly relevant model for such a use case, although it worth noting that all participants in study were able to speak). Furthermore, the authors do a very good job of motivating the work, in particular by pointing out why decoding purely motor representations of attempted or imagined speech (as has been the dominant strand of recent research) may not work well in patients whose injuries/disease affects those cortical areas. For such patients, going "upstream" to more diffuse linguistic representations will be important, and this study includes recordings from such areas in addition to traditional speech sensorimotor areas.

The manuscript includes data from nine participants across three different sites/groups. This kind of data sharing / combination of precious human data is highly commendable, as it is consistent with maximizing the value of clinical trial participants' time and risk in order to accelerate the pace of translational research.

The manuscript is very well written; its motivations are clear, the methods and results are explained clearly and succinctly, etc. The Discussion also does a good job of pointing out how the lessons learned can be applied to future efforts to develop speech BCIs.

The finding that there was speech-related information in CFC in multiple brain areas is useful and suggests further investigation of these signals may be fruitful.

The many supplementary figures, which show the same analyses as in main figures but broken down across the three studies, are appreciated and help dispel the concern that differences in strategies (For example between "imagine hearing" in studies 1 and "imagine saying" in studies 2 and 3) would be lost when aggregating across studies as in the main figures.

STUDY LIMITATIONS:

MAJOR:

This manuscript is framed as a BCI-focused design study investigating which recordings locations, neural features, and speech representations are better for decoding imagined speech. One of the key results is that the imagined speech correlations and decoding performance is low (above chance, but not by much, even when decoding between just two classes, as shown in Fig 7). This is a far cry from recent overt speech synthesis and/or classification results (which the authors do point out to highlight how decoding imagined speech appears to be a much harder problem). I recognize that the goal of the study is not to demonstrate a high performing speech BCI, but rather to make progress in exploring the design space for imagined speech BCI. However, with this goal in mind, it is unclear if the lessons learned from the reported very low SNR data will generalize to that future imagined speech BCIs that will likely need to record a different type of signal (or at least with much higher density and/or coverage). This is not to say that the present results are not useful, but I do think that their applicability for future speech BCIs is constrained by the very low SNR of these signals.

MEDIUM-LEVEL:

The manuscript reports that one of the biggest differences between overt and imagined seems to be in the BCA band; this is also the band where one would be most worried about microphonic artifact (Roussel et al 2020, which is ref 47 in this manuscript), which is of course absent during imagined speech (as the authors point out in the Discussion). The manuscript states "To enable a fair comparison of overt and imagined speech in our study, we took care of checking that the three current datasets were free of acoustic contamination". However, no data/analyses are presented. Given that the lack of artifact is critical to believing that this major difference is neural and not artifactual, it would be helpful if the authors present results (in a Supp Fig) that show there is not acoustic contamination.

I'd also have liked to see some presentation of how the audio/video data "confirmed that they did not silently mouth or whisper words", though I consider this a less important item.

MINOR:

Several figure captions have a variant of "only significant electrodes are shown". It would be helpful if this included something like "(only significant electrodes are shown, X/Y)" where X = the number of significant electrodes and Y = the total number of electrodes. This would help orient the reader to how rare (or not) the significant electrodes shown are.

Typo: extra hyphen on line 34 "and-decode"

In Figure 7, panel a has an ever-so-slightly different y range than panel b (0.3-0.8 vs 0.3-0.7). This is a small enough difference that I worry it can be missed by an inattentive reader (and comparing power features decoding vs phase-amplitude features decoding is part of the main message). Given how similar these ranges are, it shouldn't compress panel b too much to just have the same range as in a.

Reviewer #2 (Remarks to the Author):

The subject of the paper is certainly interesting and topical. In particular, the focus on imagined speech is important to counter-balance a lot of the literature focused on decoding overt speech or listening responses to speech directly from neural activity. It is encouraging to see this type of investigation being undertaken.

An important aspect is the discrimination through articulatory, phonetic, vocalic, and semantic representations but there is no definitive conclusion on which is ultimately the best for imagined speech. Findings indicate high-frequency activity corresponds to greater over speech decoding

while both high and low frequency components contributed to imagined speech decoding. The authors suggest that cross-frequency dynamics contain information for imagined speech decoding.

Overall, the manuscript is well written. However, there are some improvements needed. The introduction presents some interesting literature on speech models and how it can inform imagined speech decoding but I think it would benefit greatly by making more direct links between the literature and the specific aims/methods of the research. It is my opinion that the results section requires more clarity around how the different study designs are integrated into the analysis.

I would be somewhat concerned about possible conflation of imagined speech with imagined hearing based on the different experiments so it is essential that this is made clear in the paper.

Does the command to imagine hearing properly reflect imagined speech? The command given is not to imagine speech but to imagine hearing – does this have confounding effect on the results when mixing dataset with different instructions given to subjects?

Below, I have included some specific comments that I hope you will find constructive:

Summary

Line 20 – should be “...approaches to decoding imagined speech have met limited success...” or “decoding imagined speech has met limited success...”

Line 20 – “weak and variable” – perhaps should be compared with something, e.g. overt speech.

Introduction

Line 34 – unnecessary hyphen; probably don’t need to have both ‘classify’ and ‘decode’

Line 38 – “articulatory motor commands produced by the brain” I don’t like this phrasing. Perhaps something like “... train algorithms on neural activity corresponding to articulatory motor commands produced during overt and...”

Line 39-42 – It is not immediately clear from reading this sentence whether you are saying that the “minority of the patients with severe speech production deficits” are the same as those with Motor Neuron Disease. – This should be clarified.

Line 44 – Might be worthwhile to specify particular regions associated with perceptual or lexical representations.

Line 45 – small grammatical change - “...alternative hypotheses requires that researchers work directly from imagined speech neural signals”

Line 42-46 – I think this section would benefit from additional citations, particularly regarding the effect of post-stroke aphasia on cortical language regions, and the different characteristics of imagined speech around variability, signal-to-noise etc.

Line 51 – EEG/MEG – don’t think these terms have been specified previously in the manuscript.

Line 52 – No information provided as to what constitutes “encouraging developments” – perhaps mentions the units of speech being decoded/paradigms/results

Line 52 – I am not sure what is meant by the term “most relevant approach” and it should be qualified. Perhaps “Given the focus of this paper...” or “The most effective approach to speech decoding is based on ECoG”

Line 54, 55 – References needed for the different conditions i.e. speak aloud, imagine listening, hearing

Line 55-57 – this sentence definitely needs a reference

Line 60 – may revise to -“...production system, BHA features have not yielded decoding accuracies for imagined or covert speech equivalent to those seen with overt speech.”

May be useful to mention some of the alternative feature representations/spaces used elsewhere, e.g. Nyguen et al. 2017/18

Line 73 “In this case, neural activity is shaped by the way each individual imagines speech” This is an important point and could be expanded upon, particularly in relation to the (possibly) different was participants may have imagined speech for the data used here. Could also be linked more directly to the final sentence of this paragraph.

Line 79 – word ‘that’ duplicated

Line 79/80 – would revise to “...suggest that frequency features other than BHA are critical to

speech...”

Line 90 – should perhaps change “the range” to “a range”.

I think a bit more information on the hypothesis-driven approach would be useful at the end of the introduction. Due to the journal format, it might help the reader in digesting the results section. A very brief summation of the findings may also be helpful at this stage.

Overall, the introduction is good, but I would like to see more linking between the literature and the specific ways in which it informed the hypotheses and methods used here. This may involve discussing a little more of the approach as suggest above.

Results

Is there analysis of how the different experiments differentially effect result?

What is the motivation for the different constructions of the experiments?

Line 123/4 – try to be specific about the ways in which power spectrum changes were not identical.

The general finding that imagined speech is weaker or exhibits less significant effects is common with the literature.

In the opening paragraph, I think it is important to specify the studies from which the results are being reported. There are significant differences between the experimental procedure that may impact the findings so it is necessary to be clear about this.

This is the case for Fig. 2 for example. If the analysis is across all experiments, please state that in the caption.

Lines 152-164 – This paragraph could benefit from specific correlation values / statistical results. “Significant correlations” are mentioned without p-values and the word “modest” is too vague.

The results pertaining to imagine speech exhibit substantially weaker effects, perhaps suggesting that they may not be the best features for decoding imagined speech. On the other hand, virtually all studies indicate similar weakness of imagined speech effects.

Results presented in page 10 are too general. Specific fisher distance/anova results linked to figures/tables would be more informative

On my reading CFC does not appear useful for decoding either overt or imagined speech.

A multiclass approach to classification could have been a more robust experiment to test decoding without losing information on the different representations

For imagined speech BHA did not yield superior score to other bands. Is this a sign that this is not a good feature for imagined speech or simply an indicator of the difficulties in obtaining any strong representations of imagined speech?

Line 244/5 – Could you expand on how the flexible abstraction hypothesis support results. I think this is important.

Some statistical analysis of the decoding performance relative to chance would be useful as the CFC results in particular seem to average close to mean based on Fig 7. This may indicate that CFC are not a very useful feature for decoding either overt or imagined speech.

Line 247/8 – any suggestions for realigning imagined speech trials as with overt?

Discussion

Line 272 – change “articulation” to “articulated”

The discussion requires some separation of the decoding results pertaining to overt and imagined speech so that the overall difference in performance/accuracy can be appreciated.

Linked to the above, I think it is essential to place the results in the context of other recent decoding studies using ECoG

It might be useful to address the potential for differences in the analysis stemming from the different types of cues used in the experiments (e.g. audio vs reading)

Would be useful to mention in the decoding results that SVM was used for classification of the feature representations.

Reviewer #3 (Remarks to the Author):

This manuscript investigated the use of low- and cross-frequency features in decoding imagined and overt (loud) speech from electrocorticography (ECoG) signals. There ECoG datasets were included, in which 13 patients performed free word, rhythmic word, and rhythmic syllabic repetition tasks. Instead of using an engineering approach, this study used hypothesis-driven

approach assuming a role of low-frequency neural oscillations and their cross-frequency coupling in speech processing, within both perceptual (phonetic and vocalic) and motor representation spaces. Thus, they selected four frequency bands and their features (i.e., theta, 4-8 hz, low beta, low-gamma, and broadband high-frequency, 80-150 hz). Experimental results indicated while high-frequency activity provided the best signal for overt speech, both low- and higher- frequency power and local cross-frequency contributed to imagined speech decoding. These findings are interesting, which provide some guidance to imagined speech decoding-based BCI. The writing is general clear. A few weaknesses, however, prevented the manuscript being published in its current form.

First, the title does not reflect the novelty of this paper. In recent literature, researchers have actually used non-invasive neural (MEG) signals to show low frequency can be (also) used to decode imagined speech already (see the references below). These studies have suggested low-frequency information (including theta and even delta waves) contained key information for imagined (and overt) speech decoding. However, this manuscript provided a deep comparison of these low frequencies and their cross-frequency coupling features in imagined and overt speech decoding using ECoG data, where the novelty includes cross-frequency coupling features, invasive data, and other analysis including articulatory, phonetic, and vocalic representations. Thus, I think a more appropriate title should solve this issue, e.g., Low- and cross-frequency features in decoding imagined and overt speech.

Second, in Introduction, a detailed review of some highly relevant but missing studies in literature is needed. These works are the most relevant (see references below). Line 51 simply cited four papers using EEG (or fNIRS-EEG) and then focused on these studies that mainly used broad high gammas (BHA) of ECoG data.

Dash, D., Ferrari, P., & Wang, J. (2020). Decoding imagined and spoken phrases from non-invasive neural (MEG) signals, *Frontiers in Neuroscience*, 14(290), 1-15.

Dash, D., Ferrari, P., & Wang, J. (2020). Role of brainwaves in neural speech decoding, *Proceedings of the 28th European Signal Processing Conference (EUSIPCO)*, Amsterdam, NL, pp. 1357 - 1361.

Dash, D., Ferrari, P., Hernandez, A., Heitzman, D., Austin, S., & Wang, J. (2020). Neural speech decoding for amyotrophic lateral sclerosis, *Proc. Interspeech*, Shanghai, China, pp. 2782 - 2786.

Dash, D., Ferrari, F., & Wang, J. (2019). Spatial and spectral fingerprint in the brain: Speaker identification from single trial MEG signals, *Proc. Interspeech*, pp. 1203-1207.

Dash, D., Wisler, A., Ferrari, P., Davenport, E., Maldjian, J., & Wang, J. (2020). MEG sensor selection for neural speech decoding, *IEEE Access*, 8, 182320-182337.

Third, the experimental design could be improved by adding delta (0-4hz). I think delta is very interesting according to the literature (but was ignored) here, likely due to that the authors were not aware of the latest advance literature in decoding imagining speech (as mentioned above). A rationale is needed if the authors do not want to add delta.

Below are relatively moderate or minor comments.

Title

The ending period should be removed.

Results

Line 119. The sub-section title may miss a word "task", as this sub-section contains "Speech task and item discrimination from....".

The caption of Figure 2 is isolated with the figure. The caption can be moved to the previous page.

I think Supp. Fig. 5 should be moved to the main text, as it contains interesting (task classification accuracies) results. It would be good to have an interpretation on supp. Fig. 5 as well. For example, it seems BHA (high frequencies) can better distinguish the two tasks (imagination vs overt). This is consistent with the hypotheses in this study.

Line 187, I think I understand why "python" and "cowboys" are assigned to the dorsal group. but the description is still not that straightforward. Some readers may think should they be assigned to two distinct groups (dorsal and a non-dorsal group, respectively)? "cowboys" has a dorsal consonant /k/, while "python" does not. Some explanation may be helpful, for example, Here the dorsal group means the words in this group has a dorsal difference (one has a dorsal sound, but the other does not).

Line 191, "each group" => "each sub-group".

Figure 5, what do the left and right pictures represent, respectively? Both the pictures have left and right hemispheres already.

Figure 5, horizontal axis labels are missing in the left picture.

Discussion

The discussion can be improved when including the missing literature as mentioned above. Some findings in this study are consistent findings with the literatures (low frequency are actually very helpful in decoding imaging speech).

Method

Line 380, it is unclear what "free" means in the free word repetition task. It is a time-locked task (common design in neuroscience). It may be good to simply called the task "word repetition" or "normal word repetition" (to distinguish from rhythmic word repetition). This is minor. It's ok not to change it.

Pairwise correlation (starting from line 494). A motivation is needed about the use of pairwise correlation of features with words, where the words were labelled to 1 and -1. What's the advantages/benefits of the correlation over (binary) classification? Classification accuracy may better represent the relations between these words and their associated features, as classification models are more powerful than correlation.

NCOMMS-21-14024
Response to Reviewers

Italics: Editor/Reviewer comments

Blue: author responses

Reviewer #1:

SUMMARY:

This study led by Proix and Saa uses electrocorticography (ECoG) recordings in people undergoing epilepsy monitoring to study the electrophysiological correlates of overt (spoken out loud) and imagined speech. The manuscript's primary focus is to characterize imagined speech (vis-à-vis its differences compared to overt speech), with the motivation that this knowledge will help inform the design of speech brain-computer interfaces (BCIs) to help patients with severe speech production impairments. The authors compare the relationship between word properties (articulatory, phonetic, and vocalic) with a number of electrical field potential features (i.e., four different frequency band powers and cross-frequency coupling between bands), in four different anatomical areas (sensorimotor, middle/inferior temporal, superior temporal, and inferior frontal). They report a number of findings including that although broadband LFP was best correlated with overt speech (consistent with a number of previous studies), for decoding imagined speech it was comparable (and both were quite poor) to lower frequency features and phase-amplitude coupling. These results provide some insight into future directions for trying to improve imaginary speech BCIs, and represent a rare and extensive collection of neural recordings in people during both overt and imaginary speaking tasks.

STUDY STRENGTHS:

The study of imagined speech is timely, as speech BCIs based on decoding overt speech have progressed rapidly in recent years, but so far there has been comparatively much less work in speech BCIs for people who cannot actually speak (and imagined speech is one highly relevant model for such a use case, although it worth noting that all participants in study were able to speak). Furthermore, the authors do a very good job of motivating the work, in particular by pointing out why decoding purely motor representations of attempted or imagined speech (as has been the dominant strand of recent research) may not work well in patients whose injuries/disease affects those cortical areas. For such patients, going "upstream" to more diffuse linguistic representations will be important, and this study includes recordings from such areas in addition to traditional speech sensorimotor areas.

The manuscript includes data from nine participants across three different sites/groups. This kind of data sharing / combination of precious human data is highly commendable, as it is consistent with maximizing the value of clinical trial participants' time and risk in order to accelerate the pace of translational research.

The manuscript is very well written; its motivations are clear, the methods and results are explained clearly and succinctly, etc. The Discussion also does a good job of pointing out how the lessons learned can be applied to future efforts to develop speech BCIs.

The finding that there was speech-related information in CFC in multiple brain areas is useful and suggests further investigation of these signals may be fruitful.

The many supplementary figures, which show the same analyses as in main figures but broken down across the three studies, are appreciated and help dispel the concern that differences in strategies (For example between “imagine hearing” in studies 1 and “imagine saying” in studies 2 and 3) would be lost when aggregating across studies as in the main figures.

We thank the reviewer for the encouraging comments and appreciation of our manuscript.

STUDY LIMITATIONS:

MAJOR:

This manuscript is framed as a BCI-focused design study investigating which recordings locations, neural features, and speech representations are better for decoding imagined speech. One of the key results is that the imagined speech correlations and decoding performance is low (above chance, but not by much, even when decoding between just two classes, as shown in Fig 7). This is a far cry from recent overt speech synthesis and/or classification results (which the authors do point out to highlight how decoding imagined speech appears to be a much harder problem). I recognize that the goal of the study is not to demonstrate a high performing speech BCI, but rather to make progress in exploring the design space for imagined speech BCI. However, with this goal in mind, it is unclear if the lessons learned from the reported very low SNR data will generalize to that future imagined speech BCIs that will likely need to record a different type of signal (or at least with much higher density and/or coverage). This is not to say that the present results are not useful, but I do think that their applicability for future speech BCIs is constrained by the very low SNR of these signals.

We thank the reviewer for this thoughtful comment about the limitation(s) of our study, which we somehow do share. Low SNR is unfortunately inherent to the use of the epileptic model for possible future cortical speech BCIs. Yet, this step is absolutely necessary. If a single first patient with aphasia was to receive cortical electrodes for a speech BCI, the clinical team would have to know where to place the electrodes (premotor, motor, sensory, temporal cortex), what type of signal to decode (high gamma vs. lower frequency activity) and what task to perform (imagine speaking vs. imagine hearing). Unlike previous studies we focused here on *imagined speech* and explored a *large variety* of spatial, spectro-temporal and linguistic features, with no a priori that one should work better than the other. Our approach is both exploratory and comparative, and we believe the conclusions are solid despite weak SNRs (reproducible across patients, statistical testing, etc.).

Our main conclusion is that broadband high-frequency activity, which is the state-of-the-art signal for *overt speech* decoding, does not lead to better decoding for *covert speech* than other features such as low-frequency and cross-frequency coupling. Our results also demonstrate that temporal/auditory regions are equally interesting to sample in addition to the more common sensory and motor cortex, and that an “imagine hearing” task could be applied in patients with acute or degenerative mutism.

We hope we convinced the reviewer of the specific relevance of our study for future real BCI studies, as it addresses a number of concrete issues that have to be dealt with prior to a dedicated cortical speech-BCI (namely where to place the electrodes, what to decode, what task to consider).

We have now added a paragraph in our discussion to introduce these limitations of our study.

MEDIUM-LEVEL:

The manuscript reports that one of the biggest differences between overt and imagined seems to be in the BCA band; this is also the band where one would be most worried about microphonic artifact (Roussel et al 2020, which is ref 47 in this manuscript), which is of course absent during imagined speech (as the authors point out in the Discussion). The manuscript states “To enable a fair comparison of overt and imagined speech in our study, we took care of checking that the three current datasets were free of acoustic contamination”. However, no data/analyses are presented. Given that the lack of artifact is critical to believing that this major difference is neural and not artifactual, it would be helpful if the authors present results (in a Supp Fig) that show there is not acoustic contamination.

I'd also have liked to see some presentation of how the audio/video data “confirmed that they did not silently mouth or whisper words”, though I consider this a less important item.

We now present in Suppl. Fig. 22 audio-neural contamination matrices for one participant of each study. As stated previously, no significant acoustic contamination was detected, as compared to surrogate contamination matrices.

To confirm that participants did not silently mouth or whisper words, three controls were performed:

1. During the experiment, the experimenter was checking visually and by ear that the participant was not whispering or silently mouthing words, and correcting him/her if necessary.
2. Recorded audio of each experiment and patient were independently examined by two of the authors of the study (TP and JD) to check that no speech could be heard during the imagined speech segment. One patient was excluded based on this ground. This is now reported in the Methods.
3. Video recordings of the participants' behavior during the tasks were screened by one of the author (TP) for mouth movements during the imagined task.

Given the small amount of patients we decided it would be inappropriate to apply automatic detection of movement and whispering. We now report these details in a dedicated paragraph in the Methods.

MINOR:

Several figure captions have a variant of “only significant electrodes are shown”. It would be helpful if this included something like “(only significant electrodes are shown, X/Y)” where X = the number of significant electrodes and Y = the total number of electrodes. This would help orient the reader to how rare (or not) the significant electrodes shown are.

We now systematically indicate in the figures the number of significant electrodes for each plot.

Typo: extra hyphen on line 34 “and-decode”

We corrected the typo.

In Figure 7, panel a has an ever-so-slightly different y range than panel b (0.3-0.8 vs 0.3-0.7). This is a small enough difference that I worry it can be missed by an inattentive reader (and comparing power features decoding vs phase-amplitude features decoding is part of the main message). Given how similar these ranges are, it shouldn't compress panel b too much to just have the same range as in a.

We have now given the same range for the y axis of both panels of Fig. 7, as well as for Supp. Fig. 20.

Reviewer #2:

The subject of the paper is certainly interesting and topical. In particular, the focus on imagined speech is important to counter-balance a lot of the literature focused on decoding overt speech or listening responses to speech directly from neural activity. It is encouraging to see this type of investigation being undertaken.

An important aspect is the discrimination through articulatory, phonetic, vocalic, and semantic representations but there is no definitive conclusion on which is ultimately the best for imagined speech. Findings indicate high-frequency activity corresponds to greater over speech decoding while both high and low frequency components contributed to imagined speech decoding. The authors suggest that cross-frequency dynamics contain information for imagined speech decoding.

Overall, the manuscript is well written. However, there are some improvements needed. The introduction presents some interesting literature on speech models and how it can inform imagined speech decoding but I think it would benefit greatly by making more direct links between the literature and the specific aims/methods of the research.

We thank the reviewer for his/her comments that helped us improve the manuscript. We have followed the reviewer's recommendation, and have now introduced more direct links in the introduction between the literature and the aims and methods of our research.

It is my opinion that the results section requires more clarity around how the different study designs are integrated into the analysis.

We apologize for the lack of clarity with regard to study design (see also our answers below). We now explicitly state at the beginning of the results section that results pooled across the three studies are reported in the main text, while each study is described separately in the supplementary figures. We hope this makes the results section clearer.

I would be somewhat concerned about possible conflation of imagined speech with imagined hearing based on the different experiments so it is essential that this is made clear in the paper.

We already provided separate results in Supplementary Figures 2-4, 6-11, and 13-15 (for each of the three studies), which we believe highlights the differences between imagined speech and imagined hearing. We now refer more explicitly to these figures in the manuscript, notably in the captions of each figure. We also address the general difference between the cortical substrates of imagining hearing vs. speaking in the Reviewer's next comment.

Does the command to imagine hearing properly reflect imagined speech? The command given is not to imagine speech but to imagine hearing – does this have confounding effect on the results when mixing dataset with different instructions given to subjects?

Our assumption was indeed that “imagine hearing” would partially tap on different circuitry than “imagine speaking”, i.e., a more fronto-temporal than fronto-parietal (sensorimotor) network. From our results, it is however difficult to draw a general conclusion, as the electrode locations were different for each study. Specifically, for study 1 (imagine hearing), electrodes were located mainly in the temporal lobe (Supp. Fig. 18, 20), for study 2 (imagine speaking), electrodes were located mainly in the sensory and motor areas (Supp. Fig 18 and 20). For study 1 and 2, we do see significant Fisher distance and thus relevant discriminant information in brain regions corresponding to our hypothesis (i.e. sensory and motor for imagined speaking vs. superior temporal lobe for imagine hearing). However, Study 3 (imagine speaking), for which there are electrodes both in temporal lobe and sensory and motor areas, does not show a fronto-parietal bias (significant Fisher distances both in sensorimotor areas and the superior temporal gyrus, only for restricted set of syllables –ba, da, ga –, see Supp. Fig. 18).

The cerebral substrates of imagining oneself speak vs. imagining hearing speech are known to overlap significantly, but there are also important differences. Imagined speaking recruits fronto-parietal sensorimotor regions more strongly, suggesting that this task involves a motor-to-sensory transformation. On the contrary, imagined hearing activates more the inferior parietal cortex and intraparietal sulcus, which could correspond to a sensory memory-retrieval operation (Tian and Poeppel, J Cogn Neurosci 2013; Tian et al., Cortex 2016). Of note, most studies on the topic did not specifically contrast imagining oneself speak vs. imagining hearing one’s own voice (Alderson-Day et al., 2015). In a single-subject fMRI study (Kühn et al., 2014), the participant activated her left inferior frontal gyrus more when she reported imagining speaking vs. hearing herself speak. Future studies with large coverage will have to tell whether imagine hearing is a real useful alternative to imagine speaking when developing speech-BCI for patients with e.g. Broca’s aphasia. Given the prevalence of post-stroke expressive aphasia worldwide, it is essential to explore all dimensions of “imagined speech”, including the hearing dimension. We now mention this important issue in the discussion.

Below, I have included some specific comments that I hope you will find constructive:

Summary

Line 20 – should be “...approaches to decoding imagined speech have met limited success...” or “decoding imagined speech has met limited success...”

We have corrected the sentence as suggested.

Line 20 – “weak and variable” – perhaps should be compared with something, e.g. overt speech.

We added the comparison with overt speech.

Introduction

Line 34 – unnecessary hyphen; probably don’t need to have both ‘classify’ and ‘decode’

We removed the extra hyphen, as well as the ‘classify’.

Line 38 – “articulatory motor commands produced by the brain” I don’t like this phrasing. Perhaps something like “... train algorithms on neural activity corresponding to articulatory motor commands produced during overt and...”

We modified as suggested.

Line 39-42 – It is not immediately clear from reading this sentence whether you are saying that the “minority of the patients with severe speech production deficits” are the same as those with Motor Neuron Disease. – This should be clarified.

We added parenthesis to clarify our sentence.

Line 44 – Might be worthwhile to specify particular regions associated with perceptual or lexical representations.

We now specify which specific regions are associated with perceptual and lexical representations, i.e. temporo-parieto-occipital junction, the superior temporal gyrus, and the ventral anterior temporal regions.

Line 45 – small grammatical change - “...alternative hypotheses requires that researchers work directly from imagined speech neural signals”

We corrected the sentence.

Line 42-46 – I think this section would benefit from additional citations, particularly regarding the effect of post-stroke aphasia on cortical language regions, and the different characteristics of imagined speech around variability, signal-to-noise etc.

We have now added new references to support this section. In particular, the article of Geva et al. is a brilliant demonstration of using lesion mapping to study imagined speech.

Line 51 – EEG/MEG – don’t think these terms have been specified previously in the manuscript.

We now define those terms.

Line 52 – No information provided as to what constitutes “encouraging developments” – perhaps mentions the units of speech being decoded/paradigms/results

We now provide information about the paradigm, the units of speech being decoded, and the results presented in these articles

Line 52 – I am not sure what is meant by the term “most relevant approach” and it should be qualified. Perhaps “Given the focus of this paper...” or “The most effective approach to speech decoding is based on ECoG”

We now follow the reviewer’s recommendation.

Line 54, 55 – References needed for the different conditions i.e. speak aloud, imagine listening, hearing

We have added the relevant references.

Line 55-57 – this sentence definitely needs a reference

References were added to support this statement.

Line 60 – may revise to -“...production system, BHA features have not yielded decoding accuracies for imagined or covert speech equivalent to those seen with overt speech.”

We changed the sentence as proposed.

May be useful to mention some of the alternative feature representations/spaces used elsewhere, e.g. Nyguen et al. 2017/18

We now cite the article of Nguyen as an example of alternative feature representation, i.e. EEG spectral features in a Riemannian space.

Line 73 “In this case, neural activity is shaped by the way each individual imagines speech” This is an important point and could be expanded upon, particularly in relation to the (possibly) different way participants may have imagined speech for the data used here. Could also be linked more directly to the final sentence of this paragraph.

We have now expanded this point in the manuscript, and have linked it more directly to the final sentence of this paragraph.

Line 79 – word ‘that’ duplicated

Removed.

Line 79/80 – would revise to “...suggest that frequency features other than BHA are critical to speech...”

We implemented the suggested style improvement.

Line 90 – should perhaps change “the range” to “a range”.

Better indeed, done.

I think a bit more information on the hypothesis-driven approach would be useful at the end of the introduction. Due to the journal format, it might help the reader in digesting the results section. A very brief summation of the findings may also be helpful at this stage.

We now summarize briefly our findings and our hypothesis-driven approach at the end of the introduction.

Overall, the introduction is good, but I would like to see more linking between the literature and the specific ways in which it informed the hypotheses and methods used here. This may involve discussing a little more of the approach as suggest above.

We have followed this and the previous suggestions, we now hope that the link between the literature and the aims and methods used here is more apparent.

Results

Is there analysis of how the different experiments differentially effect result?

As discussed above, yes, we indeed provided these results as Supplementary Figures.

What is the motivation for the different constructions of the experiments?

Experiments were designed and run independently in three different centers, and only united together in this study. We believe the diversity of the approaches is a plus as we can show generalizable findings. It also allows us to explore the potential value of imagine hearing as compared to subarticulation, which as we already pointed out has important implications for future applications in aphasia. As answered above, we found such a difference when comparing study 1 and 2, but could not confirm that it was not simply due to the different coverage of the electrodes (Supp. Fig. 18 and 20). This point is now presented in the discussion.

Line 123/4 – try to be specific about the ways in which power spectrum changes were not identical.

We modified and extended the sentence following this one to better state the power spectrum changes.

The general finding that imagined speech is weaker or exhibits less significant effects is common with the literature.

We agree and now cite more broadly the relevant literature.

In the opening paragraph, I think it is important to specify the studies from which the results are being reported. There are significant differences between the experimental procedure that may impact the findings so it is necessary to be clear about this.

This is the case for Fig. 2 for example. If the analysis is across all experiments, please state that in the caption.

All figures presented in the main text come from pooling together results from all experiments. Results for individual studies are provided as supplementary figures. We now systematically mention this in the caption of the figures of the main text.

Lines 152-164 – This paragraph could benefit from specific correlation values / statistical results. “Significant correlations” are mentioned without p-values and the word “modest” is too vague. The results pertaining to imagine speech exhibit substantially weaker effects, perhaps suggesting that they may not be the best features for decoding imagined speech. On the other hand, virtually all studies indicate similar weakness of imagined speech effects.

All electrodes displayed on the brain are significant (permutation tests, $p < 0.05$, as already mentioned in the legend of Figure 4). We have now clarified this point in the legends of all figures. We also added the number of significant electrodes, the median and IQR to give the reader a quantification of our statement, as well as the number of significant electrodes for each case of each figure.

We agree with the reviewer that results reported in the literature about covert speech indeed show weak effects in general. These results were mostly obtained using BHA, implicitly considering that it was a good feature for covert speech decoding. In our manuscript, we instead point to the

fact that BHA might not be the best feature for imagined speech decoding, and explore other features, in line with previous neurolinguistic theories. Indeed, we find features that work equally well for overt and imagined speech decoding, opening new perspectives for the future of speech BCI research, as already highlighted in the discussion. Yet, it remains clear that other features not explored here might also work, or perhaps even better. We do hope that our manuscript will encourage our colleagues to explore new paths for speech decoding.

Results presented in page 10 are too general. Specific fisher distance/anova results linked to figures/tables would be more informative

We now provide median and IQR of significant Fisher distances to support each of our statements.

On my reading CFC does not appear useful for decoding either overt or imagined speech.

We have a different reading of our results. As stated in the main text, 11 and 12 participant-representation pairs were significantly decoded for overt and covert speech respectively using CFC, to compare with the 11 participant-representation-pairs that are significant to decode covert speech with power features. We now also provide the median and IRQ decoding performances for significant participant-representation pairs. In Fig. 7, we also tried to provide new elements showing that CFC can also bring some decoding performance when used as a feature. We now modified Fig. 7 to better highlight this result.

A multiclass approach to classification could have been a more robust experiment to test decoding without losing information on the different representations

We thank the reviewer for the suggestion. However, due to the low number of trials for each word (7-24), ~~and therefore representations~~, grouping the speech items together to form a binary classification task was necessary to prevent overfitting.

For imagined speech BHA did not yield superior score to other bands. Is this a sign that this is not a good feature for imagined speech or simply an indicator of the difficulties in obtaining any strong representations of imagined speech?

This is an excellent question. We favor the first option, i.e. that this result indicate that BHA is not a good feature for imagined speech. For overt speech, the sensory and motor activity correspond to localized somatotopic BHA activity in specific regions of the sensory and motor areas. The rest of language-related activity, however, is reflected in a more diffuse network that becomes apparent in the lower frequency bands. Indeed, lower frequency bands perform equally well for overt and covert speech, with sensory-motor activity being absent for the latter. We have added this point to the discussion

Line 244/5 – Could you expand on how the flexible abstraction hypothesis support results. I think this is important.

We have now expanded the sentence to better describe what we mean.

Some statistical analysis of the decoding performance relative to chance would be useful as the CFC results in particular seem to average close to mean based on Fig 7. This may indicate that CFC are not a very useful feature for decoding either overt or imagined speech.

In Fig. 7, all points which are not transparent are significantly above chance level. We now state this more clearly in the legend of the figure. Also, the number of significant participant-representation pairs are indicated in the main text. We now also provide the median and IQR decoding performances for significant participant-representation pairs. As discussed above, we think our results show, on the contrary, that CFC is a useful feature for decoding imagined speech, even though they are clearly outperformed by power spectrum features for *overt speech* classification tasks.

Line 247/8 – any suggestions for realigning imagined speech trials as with overt?

An approach that has been used in the literature is Dynamic Time Warping (DTW) to realign the covert speech trials based on the neural activity (Martin et al., *Sci. Rep.*, 2016), as already mentioned in the Discussion. We tried a similar approach, using generalizations of the DTW to an ensemble of trials (Petitjean et al., *Pattern Recognition*, 44(3):678-693; Morel et al., 2018, *Pattern Recognition*, 74:77-89). As this approach was not successful when applied to one patient of the dataset, we did not pursue this path further, but the issue remains open and will have to be investigated in future work. This problem also emphasizes the interest of using low-frequency features that are less sensitive to the realignment issue. We now further expand on this important issue in the discussion.

Discussion

Line 272 – change “articulation” to “articulated”

We corrected this typo.

The discussion requires some separation of the decoding results pertaining to overt and imagined speech so that the overall difference in performance/accuracy can be appreciated.

We have now better separated the overt and imagined speech results in two separated paragraphs in the discussion.

Linked to the above, I think it is essential to place the results in the context of other recent decoding studies using ECoG

We now discuss our results in the context of recent overt and imagined speech intracranial EEG studies.

It might be useful to address the potential for differences in the analysis stemming from the different types of cues used in the experiments (e.g. audio vs reading)

The two studies using auditory cues (studies 2 and 3) are those where participants were asked to imagine speaking. It is therefore difficult to contrast the cue effect with that in study 1 where visual cues were used and participants asked to imagine hearing. In each dataset, specific activations of relevant brain areas for hearing or reading were probed when cues were triggered (i.e. before the overt or imagined speech analysis window), and used as a first data validation. Since it was of no direct relevance to the study of overt versus imagined speech, we did not report these results. We nevertheless now mention it as a potential confound across the studies in the discussion.

Would be useful to mention in the decoding results that SVM was used for classification of the feature representations.

We added it to the relevant discussion section.

Reviewer #3:

This manuscript investigated the use of low- and cross-frequency features in decoding imagined and overt (loud) speech from electrocorticography (ECoG) signals. There ECoG datasets were included, in which 13 patients performed free word, rhythmic word, and rhythmic syllabic repetition tasks. Instead of using an engineering approach, this study used hypothesis-driven approach assuming a role of low-frequency neural oscillations and their cross-frequency coupling in speech processing, within both perceptual (phonetic and vocalic) and motor representation spaces. Thus, they selected four frequency bands and their features (i.e., theta, 4-8 hz, low beta, low-gamma, and broadband high-frequency, 80-150 hz). Experimental results indicated while high-frequency activity provided the best signal for overt speech, both low- and higher- frequency power and local cross-frequency contributed to imagined speech decoding. These findings are interesting, which provide some guidance to imagined speech decoding-based BCI. The writing is general clear. A few weaknesses, however, prevented the manuscript being published in its current form.

First, the title does not reflect the novelty of this paper. In recent literature, researchers have actually used non-invasive neural (MEG) signals to show low frequency can be (also) used to decode imagined speech already (see the references below). These studies have suggested low-frequency information (including theta and even delta waves) contained key information for imagined (and overt) speech decoding. However, this manuscript provided a deep comparison of these low frequencies and their cross-frequency coupling features in imagined and overt speech decoding using ECoG data, where the novelty includes cross-frequency coupling features, invasive data, and other analysis including articulatory, phonetic, and vocalic representations. Thus, I think a more appropriate title should solve this issue, e.g., Low- and cross-frequency features in decoding imagined and overt speech.

We have changed the title of the article to better reflect the novelty of the study relative to the existing literature. The new title is now: "Imagined speech can (also) be decoded from low- and cross-frequency intracranial EEG features"

Second, in Introduction, a detailed review of some highly relevant but missing studies in literature is needed. These works are the most relevant (see references below). Line 51 simply cited four papers using EEG (or fNIRS-EEG) and then focused on these studies that mainly used broad high gammas (BHA) of ECoG data.

*Dash, D., Ferrari, P., & Wang, J. (2020). Decoding imagined and spoken phrases from non-invasive neural (MEG) signals, *Frontiers in Neuroscience*, 14(290), 1-15.*

*Dash, D., Ferrari, P., & Wang, J. (2020). Role of brainwaves in neural speech decoding, *Proceedings of the 28th European Signal Processing Conference (EUSIPCO)*, Amsterdam, NL, pp. 1357 - 1361.*

*Dash, D., Ferrari, P., Hernandez, A., Heitzman, D., Austin, S., & Wang, J. (2020). Neural speech decoding for amyotrophic lateral sclerosis, *Proc. Interspeech, Shanghai, China*, pp. 2782 - 2786.*

Dash, D., Ferrari, F., & Wang, J. (2019). Spatial and spectral fingerprint in the brain: Speaker identification from single trial MEG signals, Proc. Interspeech, pp. 1203-1207.

Dash, D., Wisler, A., Ferrari, P., Davenport, E., Maldjian, J., & Wang, J. (2020). MEG sensor selection for neural speech decoding, IEEE Access, 8, 182320-182337.

We apologize for missing out these important recent papers. We have now added MEG and the corresponding references to our introduction.

Third, the experimental design could be improved by adding delta (0-4hz). I think delta is very interesting according to the literature (but was ignored) here, likely due to that the authors were not aware of the latest advance literature in decoding imagining speech (as mentioned above). A rationale is needed if the authors do not want to add delta.

We thank the reviewer for this suggestion. We have voluntarily omitted the delta band because of the particular design of our experiments. Indeed, we only use short words and syllables in our sentences, but no sentences. Delta has been hypothesized to have a chunking function at the sentence and group of syllables level, which was mostly not relevant in our case, as opposed to, for instance, Dash et al., Front. Neuro., 2020.

Below are relatively moderate or minor comments.

Title

The ending period should be removed.

Done.

Results

Line 119. The sub-section title may miss a word "task", as this sub-section contains "Speech task and item discrimination from....".

We have added the missing word.

The caption of Figure 2 is isolated with the figure. The caption can be moved to the previous page.

Thank you for this comment, however we believe the formatting (i.e. figure and legend positions) will ultimately be set by the journal editor. We nevertheless move the figure close to the corresponding caption in this new manuscript version.

I think Supp. Fig. 5 should be moved to the main text, as it contains interesting (task classification accuracies) results. It would be good to have an interpretation on supp. Fig. 5 as well. For example, it seems BHA (high frequencies) can better distinguish the two tasks (imagination vs overt). This is consistent with the hypotheses in this study.

We have now moved Supp. Fig. 5 back to the main text in Fig. 2. Indeed, BHA was the most performant feature for distinguishing our tasks, which is in line with our other results. We now stress this point in the corresponding Results section.

Line 187, I think I understand why “python” and “cowboys” are assigned to the dorsal group. but the description is still not that straightforward. Some readers may think should they be assigned to two distinct groups (dorsal and a non-dorsal group, respectively)? “cowboys” has a dorsal consonant /k/, while “python” does not. Some explanation may be helpful, for example, Here the dorsal group means the words in this group has a dorsal difference (one has a dorsal sound, but the other does not).

We apologize for the lack of clarity of the explanation. We reformulated it, we hope it is clearer now.

Line 191, “each group” => “each sub-group”.

In our taxonomy, we decided to use the word “group” to refer to a class of phonemes, for instance the dorsal group, within a representation, here articulatory. In Supp. Fig. 17, we indeed show each of those groups separated. For consistency, should the reviewer allow it, we would prefer to stick to our initial taxonomy.

Figure 5, what do the left and right pictures represent, respectively? Both the pictures have left and right hemispheres already.

We apologize for the confusion. Fig. 5 is organized in two columns, with the left one for overt and the right one for covert. We now better stress this in the figure.

Figure 5, horizontal axis labels are missing in the left picture.

Thanks, we added the missing labels.

Discussion

The discussion can be improved when including the missing literature as mentioned above. Some findings in this study are consistent findings with the literatures (low frequency are actually very helpful in decoding imaging speech).

We have added the missing literature in our discussion as well.

Method

Line 380, it is unclear what “free” means in the free word repetition task. It is a time-locked task (common design in neuroscience). It may be good to simply called the task “word repetition” or “normal word repetition” (to distinguish from rhythmic word repetition). This is minor. It’s ok not to change it.

We change the subsection’s title as suggested.

Pairwise correlation (starting from line 494). A motivation is needed about the use of pairwise correlation of features with words, where the words were labelled to 1 and -1. What's the advantages/benefits of the correlation over (binary) classification? Classification accuracy may better represent the relations between these words and their associated features, as classification models are more powerful than correlation.

We thank the reviewer for giving us the opportunity to better motivate the choice of analyses for the study. One can think of correlation as an encoding model (i.e. checking the correspondence between neural activity and stimuli), while classification correspond to a decoding model (predicting the stimuli from neural activity). Therefore, the two analyses serve different purposes and bring distinct answers, for instance the correlation model brings information about the brain regions involved into encoding covert speech, an information which is not directly given by decoding. We hope this clarify the role of our analyses. We now highlight this point in our manuscript.

REVIEWERS' COMMENTS

Reviewer #1 (Remarks to the Author):

The authors have thoroughly addressed the several technical or clarification questions I posed for the original manuscript. It also appears that they've comprehensively responded to the other reviewers' feedback – I think the manuscript is in good shape and this is a very sound study, well described and executed.

In terms of my major question of just how informative the present study is for guiding future cortical speech BCIs: I appreciate the added discussion paragraph, and I do agree that there's value in pointing out that decoding imagined speech may benefit from a broader search in terms of cortical areas to record from and signal features to use (e.g., the low-frequency neural features and cross-frequency coupling as emphasized here). I still believe that this "design roadmap" framing of the manuscript would have pointed a more clear path if the present results led to higher offline performance, but this is not the author's "fault" – the data are what they are (low SNR) and this is going to be a challenge for this field. This manuscript highlights this and points to possible approaches which can hopefully be built upon towards higher performance. Time will tell how much of "promising avenue" these directions will prove for achieving high performing imagined speech BCIs.

Reviewer #2 (Remarks to the Author):

The authors have taken significant steps to address the majority of my comments, and have provided sufficient justification when disagreeing with or challenging my queries. Additional clarifications, discussion and relevant citations have now been included to enhance readability and story of this work. I hope that my feedback has helped improve the overall impact of the manuscript and the important discussion points.

Reviewer #3 (Remarks to the Author):

The authors had addressed all my comments and the quality of this manuscript has been significantly improved. I have no further comments on the revision, hence, recommend it for publication.

There seems no copy of the manuscript with changes marked or highlighted, which causes inconvenience in the reviewing the revision - needed a lot of time to found out where and what have been changed.